# Pseudo-Private Data Guided Model Inversion Attacks

**Xiong Peng**[1]    **Bo Han**[1][†]    **Feng Liu**[2]    **Tongliang Liu**[3]    **Mingyuan Zhou**[4]

[1]TMLR Group, Department of Computer Science, Hong Kong Baptist University
[2]School of Computing and Information Systems, The University of Melbourne
[3]Sydney AI Centre, The University of Sydney
[4]McCombs School of Business, The University of Texas at Austin

{csxpeng, bhanml}@comp.hkbu.edu.hk
fengliu.ml@gmail.com    tongliang.liu@sydney.edu.au    mingyuan.zhou@mccombs.utexas.edu

## Abstract

In *model inversion attacks* (MIAs), adversaries attempt to recover the private training data by exploiting access to a well-trained target model. Recent advancements have improved MIA performance using a two-stage generative framework. This approach first employs a *generative adversarial network* to learn a fixed distributional prior, which is then used to guide the inversion process during the attack. However, in this paper, we observed a phenomenon that such a *fixed prior* would lead to a low probability of sampling actual private data during the inversion process due to the inherent distribution gap between the prior distribution and the private data distribution, thereby constraining attack performance. To address this limitation, we propose increasing the density around high-quality pseudo-private data—recovered samples through model inversion that exhibit characteristics of the private training data—by slightly tuning the generator. This strategy effectively increases the probability of sampling actual private data that is close to these pseudo-private data during the inversion process. After integrating our method, the generative model inversion pipeline is strengthened, leading to improvements over state-of-the-art MIAs. This paves the way for new research directions in generative MIAs. Our source code is available at: `https://github.com/tmlr-group/PPDG-MI`.

## 1 Introduction

Currently, *machine learning* (ML) models, especially *deep neural networks* (DNNs), have become prevalent in privacy-sensitive applications such as secure systems [Yin et al., 2020], personal chatbots [Ouyang et al., 2022] and healthcare services [Murdoch, 2021]. These applications inevitably rely on private and confidential datasets during model training, raising concerns about potential privacy leakages [Liu et al., 2021]. Unfortunately, recent studies reveal that ML models are vulnerable to various privacy attacks [Fredrikson et al., 2014, Krishna et al., 2019, Choquette-Choo et al., 2021]. *Model inversion attacks* (MIAs), a category of these attacks, pose significant privacy risks, which aim to infer and recover original training data by exploiting access to a well-trained model.

In the pioneering work [Fredrikson et al., 2015], MIAs were formulated as a gradient-based optimization problem in the raw data space. The goal was to seek synthetic features that maximize the prediction score for a targeted class under the target model, exploiting the strong dependency between inputs and labels established during training. For example, in attack scenarios where the target model is a facial recognition model trained on private facial images, traditional MIAs would optimize over the synthetic images to maximize the prediction score for a target identity.

---

[†]Correspondence to Bo Han (bhanml@comp.hkbu.edu.hk).

38th Conference on Neural Information Processing Systems (NeurIPS 2024).

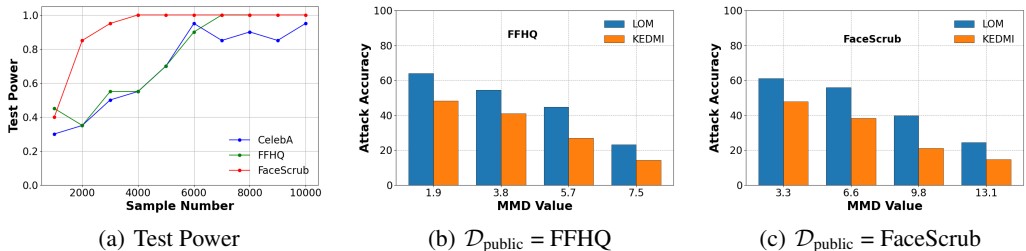

| (a) Test Power | (b) $\mathcal{D}_{\text{public}}$ = FFHQ | (c) $\mathcal{D}_{\text{public}}$ = FaceScrub |

Figure 1: **Impact of distribution discrepancies on MIAs.** (a) The test power of maximum mean discrepancy (MMD) test increases with the sample number, indicating significant differences between the distributions of $\mathcal{D}_{\text{private}}$ (CelebA) and $\mathcal{D}_{\text{public}}$ (CelebA, FFHQ and FaceScrub). (b) & (c) The proxy public datasets $\mathcal{D}'_{\text{public}}$ are crafted using the method outlined in Eq. (4). The attack performance consistently diminishes as the discrepancy between the $\mathcal{D}_{\text{private}}$ (CelebA) and $\mathcal{D}'_{\text{public}}$ increases. For detailed setups and additional results of the motivation-driven experiments, refer to Appx. C.6.

However, when the target models are DNNs, and the private features to be reconstructed reside in high-dimensional and continuous data spaces (*e.g.,* facial images), the direct optimization in the input space without any constraints is substantially ill-posed. Traditional MIAs could easily produce semantically meaningless adversarial examples [Szegedy et al., 2014], which nevertheless achieve high prediction scores under DNNs. Zhang et al. [2020] addressed this problem by employing a *generative adversarial network* (GAN) [Goodfellow et al., 2014, Radford et al., 2016] to learn a distributional prior, subsequently constraining the attack optimization space to a meaningful manifold during the inversion process. This methodology, called generative model inversion (MI), lays the groundwork for more effective model inversion of DNNs trained on high-dimensional data [Chen et al., 2021, Wang et al., 2021a, Kahla et al., 2022, Struppek et al., 2022, Nguyen et al., 2023].

Generative MIAs have shown marked improvements by incorporating a *fixed prior* in the inversion pipeline (*cf.* left panel of Fig. 2). However, this approach is fundamentally limited due to the inherent distribution discrepancy between the prior distribution and the unknown private training data distribution (*cf.* Fig. 1(a)). This discrepancy arises because the public auxiliary dataset, used to learn the distributional prior, does not intersect in labels with the private training dataset. Therefore, there is a low probability that the original private training data can be accurately sampled during the inversion process, leading to suboptimal attack performance (*cf.* Figs. 1(b) and 1(c)).

Thus, we raise a critical research question: How can the discrepancy between the prior distribution and the unknown private training data distribution be mitigated? Addressing this distribution gap is challenging in the MI context, where we only have access to a well-trained target model. Nevertheless, the target model still encapsulates information about the private training data. By performing model inversion on the target model, we can generate what we called *pseudo-private data*, which are reconstructed samples that reveal the characteristics of the private training data and can serve as its surrogate. Consequently, enhancing the density of pseudo-private data under the prior distribution indirectly increases the density of the private training data. This, in turn, raises the probability of accurately sampling the actual private training data (*cf.* right panel of Fig. 2).

To this end, we propose a novel model inversion methodology, termed *pseudo-private data guided MI (PPDG-MI)*. The efficacy of PPDG-MI is demonstrated through a simple example using a 2D dataset (Sec. 3.3), where we increase the density of the pseudo-private data by directly minimizing the distribution discrepancy between the prior distribution and empirical pseudo-private data distribution, as measured by conditional transport [Zheng and Zhou, 2021] that is amenable to mini-batch based optimization and straightforward to implement. For the density enhancement of high-dimensional data, we introduce a nuanced tuning strategy involving three iterative steps: ① Conduct a round of MIAs to produce pseudo-private samples. ② Select high-quality pseudo-private samples based on prediction scores. ③ Fine-tune the generator to increase the density around these high-quality samples, thereby increasing the probability of sampling the original private training data.

In summary, our contributions and findings are as follows:

- Conceptually, we identify a fundamental limitation common to state-of-the-art (SOTA) generative MIAs [Zhang et al., 2020, Chen et al., 2021, Kahla et al., 2022, Struppek et al., 2022, Han et al., 2023, Nguyen et al., 2023], *i.e.,* the utilization of a fixed prior during the

inversion process. We argue that this approach is sub-optimal for MIAs and introduce a novel strategy, termed *pseudo-private data guided* MI, to mitigate this limitation, thereby paving the way for future research and advancements in generative MIAs.

- Technically, we provide multiple implementations of PPDG-MI to validate the effectiveness of our proposed strategy. For low-resolution MIAs, we introduce PPDG-vanilla. For more complex high-dimensional MIAs, we offer PPDG-PW, which employs point-wise tuning, and two batch-wise tuning strategies: PPDG-MI with conditional transport (PPDG-CT) and PPDG-MI with maximum mean discrepancy (PPDG-MMD) (Sec. 3).

- Empirically, through extensive experimentation, we demonstrate that our solution significantly improves the performance of the SOTA MI methods across various settings, including white-box, black-box, and label-only MIAs (Sec. 4). Our findings emphasize the increasing risks associated with MIAs and further highlight the urgent need for more robust defenses against the leakage of private information from DNNs.

## 2 Problem Setup and Preliminary

### 2.1 Model Inversion Attacks

**Problem Setup.** Let $\mathcal{X} \subset \mathbb{R}^{d_X}$ be the feature space, and $\mathcal{Y}_{\text{private}} = \{1, \ldots, C\}$ be the private label space. The *target model*, $\mathrm{M} \colon \mathcal{X} \to [0,1]^C$, is a classifier well-trained on the private training dataset $\mathcal{D}_{\text{private}}$ sampled from $\mathrm{P}(\mathcal{X}_{\text{private}}, \mathcal{Y}_{\text{private}})$. In standard settings, for a specific class $y$ in $\mathcal{Y}_{\text{private}}$, MIAs aim to reconstruct synthetic samples by exploiting access to the target model $\mathrm{M}$ to uncover sensitive features of class $y$. In this context, the adversary is limited to querying $\mathrm{M}$, and also possesses knowledge of the target data domain but lacks specific details about $\mathcal{D}_{\text{private}}$.

Mathematically, MI is formulated as an optimization problem: Given a target class $y$, the goal is to find a sample $\mathbf{x}$ that maximizes the model $\mathrm{M}$'s prediction score for class $y$. In high-dimensional data settings, traditional MIAs [Fredrikson et al., 2015] use direct input space optimization, often leading to adversarial samples [Szegedy et al., 2014] that, despite high prediction scores, lack meaningful features. To mitigate this issue, Zhang et al. [2020] propose a generative MI approach, which learns a distributional prior to constrain the optimization to a low-dimensional, meaningful manifold.

Current generative MIAs primarily concentrate on either the initial training process of GANs [Chen et al., 2021, Yuan et al., 2023, Nguyen et al., 2024] or the optimization techniques used in the attacks [Zhang et al., 2020, Wang et al., 2021a, Struppek et al., 2022, Kahla et al., 2022, Nguyen et al., 2023]. In this paper, we take another direction and introduce a novel approach by fine-tuning the GAN's generator based on the attack results from previous runs. This method introduces a dynamic and iterative dimension to model inversion attacks, expanding the current understanding and research direction of generative MIAs. For detailed related work, please refer to Appx. A.1.

Specifically, the generative MI approach consists of two stages. Initially, a GAN learns a prior from public auxiliary datasets, in which $\mathcal{Y}_{\text{public}} \cap \mathcal{Y}_{\text{private}} = \emptyset$. This process involves a generator, denoted as $\mathrm{G}(\cdot; \boldsymbol{\theta}) \colon \mathcal{Z} \to \mathcal{X}_{\text{prior}}$, parameterized by $\boldsymbol{\theta}$, that transforms a low-dimensional latent code, $\mathbf{z} \in \mathcal{Z}$, into a high-dimensional image, $\mathbf{x} \in \mathcal{X}_{\text{prior}}$. Concurrently, a discriminator $\mathrm{D}(\cdot; \boldsymbol{\phi}) \colon \mathcal{X} \to \mathbb{R}$, which can distinguish between generated and real images. Subsequently, the MI optimization can be constrained to the latent space $\mathcal{Z}$ of the *fixed prior* $\mathrm{G}$, which can be formulated as:

$$\mathbf{z}^* = \arg\min_{\mathbf{z}} \ \mathcal{L}_{\text{id}}(\mathbf{z}; y, \mathrm{M}, \mathrm{G}) + \lambda \mathcal{L}_{\text{prior}}(\mathbf{z}; \mathrm{G}, \mathrm{D}), \tag{1}$$

where $\mathcal{L}_{\text{id}}(\cdot)$ denotes the identity loss, *e.g.*, the cross-entropy loss $-\log \mathbb{P}_{\mathrm{M}}(y|\mathrm{G}(z))$, which optimizes for an optimal synthetic sample $\mathbf{x}^* = \mathrm{G}(\mathbf{z}^*)$. Additionally, $\mathcal{L}_{\text{prior}}(\cdot)$ serves as a regularizer for the latent code $\mathbf{z}$, and the parameter $\lambda$ balances the trade-off between the identity loss and the regularizer.

### 2.2 Distribution Discrepancy Measure

To effectively align distributions in our methods, it is essential to introduce metrics that can accurately quantify the differences between them. Two commonly used measures for this purpose are *maximum mean discrepancy* (MMD) and *conditional transport* (CT). MMD focuses on mean differences

using kernel methods, while CT incorporates cost-based transport distances, offering complementary perspectives on distributional discrepancies. This section introduces the empirical estimation of MMD and CT. For more details on these discrepancy measures, please refer to Appx. A.2.

**Estimation of MMD.** Given distributions P and Q, and sample sets $S_X = \{\mathbf{x}_i\}_{i=1}^n \sim$ P and $S_Y = \{\mathbf{y}_j\}_{j=1}^m \sim$ Q, MMD can be estimated with the following estimator [Gretton et al., 2012b]:

$$\widehat{\text{MMD}}_u^2(S_X, S_Y; k) = \frac{1}{n(n-1)} \sum_{i=1}^n \sum_{j=1, j \neq i}^n k(\mathbf{x}_i, \mathbf{x}_j) + \frac{1}{m(m-1)} \sum_{i=1}^m \sum_{j=1, j \neq i}^m k(\mathbf{y}_i, \mathbf{y}_j)$$
$$- \frac{2}{mn} \sum_{i=1}^n \sum_{j=1}^m k(\mathbf{x}_i, \mathbf{y}_j). \tag{2}$$

where $k$ is a kernel function, $\mathbf{x}_i, \mathbf{x}_j \in S_X$ and $\mathbf{y}_i, \mathbf{y}_j \in S_Y$.

**Estimation of CT.** Similarly, for sample sets $S_X = \{\mathbf{x}_i\}_{i=1}^n$ and $S_Y = \{\mathbf{y}_j\}_{j=1}^m$, the CT measure can be approximated as follows [Zheng and Zhou, 2021]:

$$\text{CT}(S_X, S_Y) = \sum_{i=1}^n \sum_{j=1}^m c(\mathbf{x}_i, \mathbf{y}_j) \left( \frac{e^{-d_\psi(\mathbf{x}_i, \mathbf{y}_j)}}{\sum_{j'=1}^m e^{-d_\psi(\mathbf{x}_i, \mathbf{y}_{j'})}} + \frac{e^{-d_\psi(\mathbf{x}_i, \mathbf{y}_j)}}{\sum_{i'=1}^n e^{-d_\psi(\mathbf{x}_{i'}, \mathbf{y}_j)}} \right). \tag{3}$$

Here, $d_\psi(\mathbf{x}, \mathbf{y})$ is a function parameterized by $\psi$ that measures the similarity between $\mathbf{x}$ and $\mathbf{y}$, and $c(\mathbf{x}, \mathbf{y})$ is a cost function that measures the distance between the points $\mathbf{x}$ and $\mathbf{y}$.

## 3 Pseudo-private Data Guided Model Inversion

This section introduces our proposed methodology, *i.e., pseudo-private data guided* MI (PPDG-MI). First, we present and discuss the critical motivation that inspires our method (Sec. 3.1). Second, we introduce the general framework of PPDG-MI (Sec. 3.2). Third, to ease understanding, we demonstrate and illustrate the rationality of our solution on a simple toy dataset (Sec. 3.3). Fourth, we present a more nuanced and detailed strategy for tuning the generator to enhance density in high-dimensional image spaces, accompanied by multiple algorithmic implementations (Sec. 3.4).

### 3.1 Motivation: Effect of Distribution Discrepancies on MIAs

Collecting public auxiliary datasets that closely resemble the private dataset remains challenging. This difficulty arises because the MI adversary lacks knowledge of specific class information, and only understands the general data domain about $P(\mathcal{X}_{\text{private}})$. Thus, we hypothesize a significant distribution discrepancy between the prior distribution $P(\mathcal{X}_{\text{prior}})$ and the private data distribution $P(\mathcal{X}_{\text{private}})$. This claim is supported by Fig. 1(a), where we quantify the distribution discrepancy between commonly adopted public auxiliary datasets and private training datasets using the MMD measure [Borgwardt et al., 2006], showcasing a substantial gap between these two distributions.

To evaluate the impact of this distribution discrepancy on MI performance, we create a series of proxy prior distributions through linear interpolation, where a mixing coefficient $\alpha \in [0, 1]$ determines the proportion of samples drawn from each distribution. Specifically, a fraction $\alpha$ of samples is drawn from $P(\mathcal{X}_{\text{prior}})$, and the remaining $(1 - \alpha)$ is drawn from $P(\mathcal{X}_{\text{private}})$. This process is represented as:

$$P(\mathcal{X}'_{\text{prior}}) = \alpha P(\mathcal{X}_{\text{prior}}) + (1 - \alpha)P(\mathcal{X}_{\text{private}}). \tag{4}$$

We apply these proxy prior distributions to constrain the MI optimization as outlined in Eq. (1). As illustrated in Figs. 1(b) and 1(c), the MI performance decreases monotonically as the MMD value between $P(\mathcal{X}'_{\text{prior}})$ and $P(\mathcal{X}_{\text{private}})$ increases. This leads us to pose a critical research question:

> *How can the discrepancy between the prior distribution and the unknown private*
> *data distribution be mitigated to enhance MI performance?*

In response to this, a revised inversion pipeline is required, wherein G is dynamically adjusted throughout the inversion process. This adjustment aims to progressively narrow the distribution gap between $P(\mathcal{X}_{\text{prior}})$ and $P(\mathcal{X}_{\text{private}})$, thereby potentially improving MI performance.

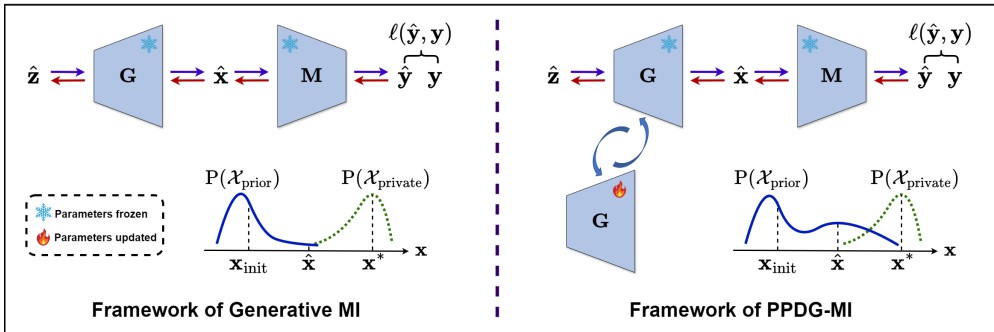

Figure 2: **Overview of traditional generative MI framework vs. *pseudo-private data guided* MI (PPDG-MI) framework.** PPDG-MI leverages pseudo-private data $\hat{\mathbf{x}}$ generated during the inversion process, which reveals the characteristics of the actual private data, to fine-tune the generator G. The goal is to enhance the density of $\hat{\mathbf{x}}$ under the learned distributional prior $P(\mathcal{X}_{\text{prior}})$, thereby increasing the probability of sampling actual private data $\mathbf{x}^*$ during the inversion process.

## 3.2 PPDG-MI Framework

This section presents a novel model inversion pipeline that dynamically adjusts the generator G to mitigate the distribution discrepancy between $P(\mathcal{X}_{\text{prior}})$ and $P(\mathcal{X}_{\text{private}})$ (*cf.* Fig. 2 for the framework overview). As aforementioned, in the MI context, while the specific details of $\mathcal{D}_{\text{private}}$ remain unknown, we have access to the target model M, which is well-trained on $\mathcal{D}_{\text{private}}$, still encapsulates information about $\mathcal{D}_{\text{private}}$. Therefore, by conducting MI on the target model M, we can generate a set of pseudo-private samples (*i.e.,* reconstructed samples), denoted as $\mathcal{D}^{\text{s}}_{\text{private}}$, which reveal the characteristics of the private dataset $\mathcal{D}_{\text{private}}$ and can serve as its surrogate. Thus, the key insight is that by enhancing the density of the prior distribution $P(\mathcal{X}_{\text{prior}})$ around $\mathcal{D}^{\text{s}}_{\text{private}}$, we indirectly increase the density around $\mathcal{D}_{\text{private}}$ as well. Consequently, the probability of sampling data from $P(\mathcal{X}_{\text{private}})$ could be increased. This strategy is termed *pseudo-private data guided* MI (PPDG-MI).

To this intuition, the proposed MI framework consists of the following three iterative steps:

**Step-1: Pseudo-private Data Generation by Conducting MI on the Target Model.** Specifically, in generative MI, optimization is restricted to the latent space $\mathcal{Z}$. Initially, we sample a set of latent codes, $\mathbf{Z} = \{\mathbf{z}_i \mid \mathbf{z}_i \in \mathcal{Z}, i = 1, \ldots, N\}$. Then, by leveraging Eq. (1), these initial latent codes are optimized to produce $\hat{\mathbf{Z}} = \{\hat{\mathbf{z}} = \arg\min \mathcal{L}_{\text{id}}(\mathbf{z}) + \lambda \mathcal{L}_{\text{prior}}(\mathbf{z}) \mid \mathbf{z} \in \mathbf{Z}\}$. Subsequently, this optimized set $\hat{\mathbf{Z}}$ is utilized to generate a pseudo-private dataset $\mathcal{D}^{\text{s}}_{\text{private}} = \{\hat{\mathbf{x}} = \mathrm{G}(\hat{\mathbf{z}}) \mid \hat{\mathbf{z}} \in \hat{\mathbf{Z}}\}$.

**Step-2: Selection of High-Quality Pseudo-private Data.** In this step, we aim to select high-quality data from $\mathcal{D}^{\text{s}}_{\text{private}}$ that closely resemble the characteristics of samples in $\mathcal{D}_{\text{private}}$, serving as their proxy. An intuitive method is to select samples with high prediction scores. Thus, following Struppek et al. [2022], we opt to select samples with larger expected prediction scores $\mathbb{E}[\mathbb{P}_{\mathrm{M}}(y \mid T(\hat{\mathbf{x}}))]$ under random image transformations $T$, indicating that $\hat{\mathbf{x}}$ represents the desired characteristics for target class $y$ more accurately. Specifically, we select a high-quality subset $\mathcal{D}^{\text{s}'}_{\text{private}}$, consisting of samples with top $K$ expected prediction scores from $\mathcal{D}^{\text{s}}_{\text{private}}$.

**Step-3: Density Enhancement around Pseudo-private Data.** In this step, we focus on fine-tuning G to adjust the prior distribution $P(\mathcal{X}_{\text{prior}})$, aiming to increase the probability of sampling data from $P(\mathcal{X}_{\text{private}})$. In the existing literature, MIAs can be categorized into two types: those targeting high-resolution tasks [Struppek et al., 2022] and those targeting low-resolution tasks [Zhang et al., 2020, Chen et al., 2021, Kahla et al., 2022, Nguyen et al., 2023]. In high-resolution MIAs, adversaries leverage *pre-trained* GANs without access to training specifics. In contrast, low-resolution MIAs involve adversaries training GANs from scratch using the public auxiliary dataset $\mathcal{D}_{\text{public}}$. This distinction enables the development of tuning strategies tailored to different attack settings.

For MIAs focusing on low-resolution tasks, where the generator G is less powerful and the low-resolution image manifold is more susceptible to disruption, we adopt a principled tuning strategy. Specifically, we fine-tune G and D using the original GAN training objective on $\mathcal{D}_{\text{public}} \cup \mathcal{D}^{\text{s}'}_{\text{private}}$, a strategy termed *PPDG-vanilla* (*cf.* Alg. 1). For MIAs focusing on high-resolution tasks, *e.g., Plug*

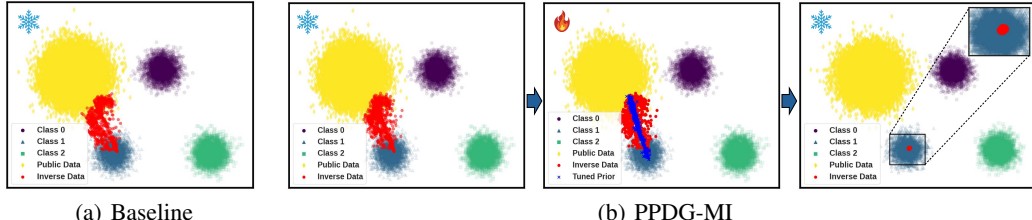

| (a) Baseline | (b) PPDG-MI |
| --- | --- |

Figure 3: **Illustration of the rationale behind PPDG-MI using a simple 2D example.** Training samples from Class 0-2 are represented by purple, blue, and green, respectively, while public auxiliary data are shown in yellow. MIAs aim to recover training samples from Class 1, with reconstructed samples shown in red. (a) Results of the baseline attack with a fixed prior. (b) Left: Pseudo-private data generation. Middle: Density enhancement of pseudo-private data under prior distribution. Right: Final attack results of PPDG-MI with the tuned prior, where all the recovered points converge to the centroid of the class distribution, indicating the most representative features are revealed.

& *Play Attacks* (PPA) [Struppek et al., 2022], we propose a tuning strategy that leverages only the high-quality pseudo-private dataset $\mathcal{D}_{\text{private}}^{\text{s}'}$, which can be formalized as follows:

$$G, D \leftarrow \texttt{Fine-tune}(G, D, \mathcal{D}_{\text{private}}^{\text{s}'}). \tag{5}$$

This adjustment aims to increase the density of the prior distribution around $\mathcal{D}_{\text{private}}^{\text{s}'}$. The concrete realizations (*cf.* Alg. 2) are presented in Sec. 3.4. After fine-tuning the generator, return to Step-1 and repeat the attack process to further improve the MI performance. Our experiments primarily focus on PPA, which allows us to investigate a more *realistic attack scenario* with high-resolution data.

### 3.3 Understanding PPDG-MI with 2D Data

To illustrate the principles of PPDG-MI, we present a toy example using a 2D dataset with three classes, each sampled from a class-conditional Gaussian distribution, as shown in Fig. 3. Additionally, a public dataset is sampled from a separate Gaussian distribution to learn the distributional prior $P(\mathcal{X}_{\text{prior}})$. We simulate a simple MIA by generating an initial set of samples using generator G and then optimizing these samples to maximize the model's prediction score for Class 1. The objective is to uncover the features of Class 1, primarily the coordinates of the training samples. The closer these optimized samples are to the centroid of Class 1's distribution, *i.e.,* the high-density region, the more effective the attack. See Appx. C.7 for a larger version of Fig. 3 and the experimental details of the toy example. An **animated illustration** of the toy demo is available in the supplementary materials.

The baseline attack results are shown in Fig. 3(a), where a fixed G is adopted during the inversion process. The left panel of Fig. 3(b) illustrates the generation of dataset $\mathcal{D}_{\text{private}}^{\text{s}}$ through a round of MI on model M. Middle panel of Fig. 3(b) shows the enhancement of density around $\mathcal{D}_{\text{private}}^{\text{s}}$ under the prior distribution $P(\mathcal{X}_{\text{prior}})$, achieved by fine-tuning $G(\cdot; \boldsymbol{\theta})$ to align with the empirical distribution of $\mathcal{D}_{\text{private}}^{\text{s}}$, using the CT measure. The final attack results of PPDG-MI are shown in the right panel of Fig. 3(b). It is evident that, in comparison to the baseline where only a small fraction of reconstructed samples fall within the high-density region of the training data distribution, all reconstructed samples from PPDG-MI are located in this high-density region. See Appx. C.7 for the quantitative results.

Although we initially applied a direct distribution match strategy in this simplified setting to implement PPDG-MI, the empirical results indicate that this approach is less effective for higher-dimensional image data, as it would destroy the generator's manifold (Appx. C.8). We address this issue by introducing a nuanced tuning strategy tailored for high-dimensional data settings, detailed in Sec. 3.4.

### 3.4 Nuanced Approach of PPDG-MI for High-Dimensional Image Data

Considering the primary baseline PPA [Struppek et al., 2022] uses StyleGAN [Karras et al., 2020] as its distributional prior, our approach leverages StyleGAN's disentangled nature, allowing slight local changes to its produced appearance without disrupting the manifold. Specifically, we first identify a high-density neighbor for each pseudo-private sample $\hat{\mathbf{x}}$ (*cf.* Fig. 4(b)) and adjust this neighbor to be closer to $\hat{\mathbf{x}}$, thereby enhancing density around $\hat{\mathbf{x}}$ in the generator's domain (*cf.* Fig. 4(c)).

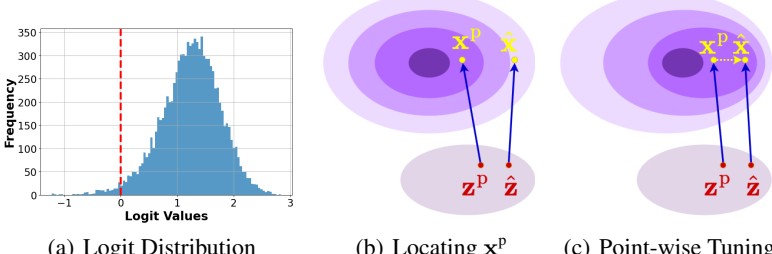

|  |  |  |
|:---:|:---:|:---:|
| (a) Logit Distribution | (b) Locating $\mathbf{x}^{\mathrm{p}}$ | (c) Point-wise Tuning |

Figure 4: **Illustration of PPDG-MI using a point-wise tuning approach.** (a) The distribution of discriminator logit outputs for randomly generated samples by the generator G, showing that the discriminator can empirically reflect the density of generated samples. (b) Locating the high-density neighbor $\mathbf{x}^{\mathrm{p}}$ by optimizing Eq. (6). Darker colors represent regions with higher density. (c) Increasing density around the pseudo-private data $\hat{\mathbf{x}}$ by moving $\mathbf{x}^{\mathrm{p}}$ towards $\hat{\mathbf{x}}$, *i.e.,* optimizing Eq. (7).

**Instantiate PPDG-MI with Point-wise Tuning.** To this intuition, we detail a two-step method to increase the density around pseudo-private samples. First, to locate a near neighbor $\mathbf{x}^{\mathrm{p}}$ of $\hat{\mathbf{x}}$, we optimize the latent code $\mathbf{z}$ to produce $\mathbf{x}^{\mathrm{p}}$. The closeness between $\mathbf{x}^{\mathrm{p}}$ and $\hat{\mathbf{x}}$ is measured using the LPIPS perceptual loss function [Zhang et al., 2018]. Additionally, to ensure that $\mathbf{x}^{\mathrm{p}}$ is located in a high-density region of $\mathrm{P}(\mathcal{X}_{\mathrm{prior}})$, we leverage the discriminator D. Although GANs typically do not provide explicit probability density functions, empirical evidence suggests that D effectively indicates the density of generated samples (*cf.* Fig. 4(a)). Overall, the optimization objective is formulated as

$$\mathbf{z}^{\mathrm{p}} = \arg\min_{\mathbf{z}} \underbrace{\mathcal{L}_{\mathrm{LPIPS}}(\hat{\mathbf{x}}, \mathrm{G}(\mathbf{z}))}_{\text{neighborhood constraint}} - \underbrace{\lambda_1 \mathrm{D}(\mathrm{G}(\mathbf{z}))}_{\text{high-density constraint}} , \tag{6}$$

where $\mathcal{L}_{\mathrm{LPIPS}}$ represents the perceptual loss function, and $\lambda_1$ is a tuning hyperparameter that balances the constraints. At this step, the generator remains frozen. After optimizing Eq. (6), we obtain a high-density neighbor point $\mathbf{x}^{\mathrm{p}} = \mathrm{G}(\mathbf{z}^{\mathrm{p}}; \boldsymbol{\theta})$ of $\hat{\mathbf{x}}$. We then aim to slightly alter $\mathrm{G}(\cdot; \boldsymbol{\theta})$ to pull $\mathbf{x}^{\mathrm{p}}$ towards $\hat{\mathbf{x}}$, thereby enhancing the local density around $\hat{\mathbf{x}}$ (*cf.* Fig. 4(c)). This is accomplished by fine-tuning the generator with the point-wise loss term:

$$\mathcal{L}_{\mathrm{PPDG-PW}}(\boldsymbol{\theta}) = \mathcal{L}_{\mathrm{LPIPS}}(\hat{\mathbf{x}}, \mathrm{G}(\mathbf{z}^{\mathrm{p}}; \boldsymbol{\theta})). \tag{7}$$

At this step, $\mathbf{z}^{\mathrm{p}}$ remains fixed, and the adjustment is applied exclusively to the generator G. Building on the point-wise density enhancement, we extend our approach to a batch-wise method using statistical distribution discrepancy measures [Borgwardt et al., 2006, Zheng and Zhou, 2021], aiming for a more principled local distribution alignment strategy.

**Instantiate PPDG-MI with Batch-wise Tuning.** Given the sets $\mathcal{D}_{\mathrm{private}}^{\mathrm{s'}} = \{\hat{\mathbf{x}}_i\}_{i=1}^{m}$ and $\{\mathrm{G}(\mathbf{z}_j)\}_{j=1}^{n} \sim \mathrm{P}(\mathcal{X}_{\mathrm{prior}})$, following the point-wise tuning settings, we initially map these samples to the LPIPS space using a feature extractor $f$. Denote $\delta$ as the distribution discrepancy measure, the batch-wise tuning strategy adapts the previous point-wise Eqs. (6) and (7) as follows:

$$\{\mathbf{z}_j^{\mathrm{p}}\}_{j=1}^{n} = \arg\min_{\{\mathbf{z}_j\}_{j=1}^{n}} \underbrace{\delta(\{f(\hat{\mathbf{x}}_i)\}_{i=1}^{m}, \{f(\mathrm{G}(\mathbf{z}_j))\}_{j=1}^{n})}_{\text{neighborhood constraint}} - \underbrace{\lambda_1 \frac{1}{n} \sum_{i=1}^{n} \mathrm{D}(\mathrm{G}(\mathbf{z}_j))}_{\text{high-density constraint}}, \tag{8a}$$

$$\mathcal{L}_{\mathrm{PPDG-BW}}(\boldsymbol{\theta}) = \delta(\{f(\hat{\mathbf{x}}_i)\}_{i=1}^{m}, \{f(\mathrm{G}(\mathbf{z}_j^{\mathrm{p}}; \boldsymbol{\theta}))\}_{j=1}^{n}). \tag{8b}$$

We present two realizations of the batch-wise tuning strategy: PPDG-MMD and PPDG-CT. When $\delta$ is set as MMD with Gaussian kernel $k$, the optimization objective in Eqs. (8a) and (8b) is realized as

$$\{\mathbf{z}_j^{\mathrm{p}}\}_{j=1}^{n} = \arg\min_{\{\mathbf{z}_j\}_{j=1}^{n}} \widehat{\mathrm{MMD}}_u^2(\{f(\hat{\mathbf{x}}_i)\}_{i=1}^{m}, \{f(\mathrm{G}(\mathbf{z}_j))\}_{j=1}^{n}; k) - \lambda_1 \frac{1}{n} \sum_{i=1}^{n} \mathrm{D}(\mathrm{G}(\mathbf{z}_j)), \tag{9a}$$

$$\mathcal{L}_{\mathrm{PPDG-MMD}}(\boldsymbol{\theta}) = \widehat{\mathrm{MMD}}_u^2(\{f(\hat{\mathbf{x}}_i)\}_{i=1}^{m}, \{f(\mathrm{G}(\mathbf{z}_j^{\mathrm{p}}; \boldsymbol{\theta}))\}_{j=1}^{n}; k). \tag{9b}$$

Similarly, when $\delta$ is set as CT, the optimization objective in Eqs. (8a) and (8b) is realized as

$$\{\mathbf{z}_j^{\mathrm{p}}\}_{j=1}^n = \underset{\{\mathbf{z}_j\}_{j=1}^n}{\arg\min} \ \mathrm{CT}(\{f(\hat{\mathbf{x}}_i)\}_{i=1}^m, \{f(\mathrm{G}(\mathbf{z}_j))\}_{j=1}^n) - \lambda_1 \frac{1}{n} \sum_{i=1}^n \mathrm{D}(\mathrm{G}(\mathbf{z}_j)), \quad (10a)$$

$$\mathcal{L}_{\mathrm{PPDG\text{-}CT}}(\boldsymbol{\theta}) = \mathrm{CT}(\{f(\hat{\mathbf{x}}_i)\}_{i=1}^m, \{f(\mathrm{G}(\mathbf{z}_j^{\mathrm{p}}; \boldsymbol{\theta}))\}_{j=1}^n). \quad (10b)$$

The cost function in Eq. (3) is implemented as $c(\mathbf{x}, \mathbf{y}) = 1 - \cos(f(\mathbf{x}), f(\mathbf{y}))$, while the distance function is implemented as $d(\mathbf{x}, \mathbf{y}) = f(\mathbf{x})^T f(\mathbf{y})$, which are commonly adopted realization choices in existing literature [Tanwisuth et al., 2021, 2023].

## 4 Experiments

In this section, we evaluate the performance of SOTA MI methods before and after integrating them with PPDG-MI, as well as the robustness against SOTA MI defenses, including BiDO [Peng et al., 2022] and NegLS [Struppek et al., 2024], to assess the overall effectiveness of PPDG-MI. The evaluation primarily focuses on real-world face recognition tasks. For high-resolution ($224 \times 224$) tasks, we consider PPA [Struppek et al., 2022] in the white-box setting. For low-resolution ($64 \times 64$) tasks, we consider GMI [Zhang et al., 2020], KEDMI [Chen et al., 2021], LOM [Nguyen et al., 2023], and PLG-MI [Yuan et al., 2023] in the white-box setting, RLB-MI [Han et al., 2023] in the black-box setting, as well as BREP-MI [Kahla et al., 2022] in the label-only setting.

### 4.1 Experimental Setup

This section briefly introduces the experimental setups. For further details, please refer to Appx. C.

**Datasets and Models.** In line with existing MIA literature on face recognition, we use the CelebA [Liu et al., 2015], FaceScrub [Ng and Winkler, 2014], and FFHQ datasets [Karras et al., 2019]. These datasets are divided into two parts: the private training dataset $\mathcal{D}_{\mathrm{private}}$ and the public auxiliary dataset $\mathcal{D}_{\mathrm{public}}$, ensuring no identity overlap. For high-resolution tasks, we trained ResNet-18 [He et al., 2016], DenseNet-121 [Huang et al., 2017] and ResNeSt-50 [Zhang et al., 2022] as target models. For low-resolution tasks, we trained VGG16 [Simonyan and Zisserman, 2015] and face.evoLVe [Wang et al., 2021b] as target models. The training details of these models are presented in Appx. C.3. We summarize the attack methods, target models, and datasets adopted in Tab. 4.

**Attack Parameters.** For all MIAs, we fine-tune the generator $\mathbf{G}$ in an identity-wise manner, to minimize alterations to the generator's latent space. Thus, adjustments to the attack parameters in official implementations are required. Detailed attack parameters are provided in Appx. C.4.

**Evaluation Metrics.** To evaluate the performance of an MIA, we need to assess whether the reconstructed images reveal private information about the target identity. Following existing literature [Zhang et al., 2020], we adopt top-1 (Acc@1) and top-5 (Acc@5) attack accuracy, as well as K-Nearest Neighbors Distance (KNN Dist). Details for these metrics are provided in Appx. C.5.

### 4.2 Main Results

In the main experiments, we integrate PPDG-MI for density enhancement while still employing the baseline attack method for MI. We conduct one round of fine-tuning of $\mathbf{G}$ and present the resulting attack results to demonstrate the efficacy of PPDG-MI. The results of multi-round fine-tuning are reserved for the ablation study (*cf.* Sec. 4.3). Additional experimental results, including evaluations on various target models, assessments with PLG-MI, black-box, and label-only MIAs, as well as comparisons against SOTA MI defenses for low-resolution tasks, are presented in Appx. D.

**Comparison with PPA in the high-resolution setting.** For each baseline setup, we report results for three variants: PPDG-PW, PPDG-CT and PPDG-MMD. The results presented in Tab. 1 demonstrate that our proposed method significantly improves MI performance across all setups, validating its effectiveness. Notably, integrating our methods with the baseline substantially increases attack accuracy. The KNN distance results also confirm that our methods more accurately reconstruct data resembling the private training data. Qualitative results of reconstructed samples from all target models are provided in Figs. 9 and 10 in Appx. D.3. Additionally, among the three PPDG-MI variants, the batch-wise tuning strategy consistently outperforms the point-wise tuning strategy. Batch-wise

Table 1: Comparison of MI performance with PPA in high-resolution settings. $\mathcal{D}_{private}$ = CelebA or FaceScrub, GANs are pre-trained on $\mathcal{D}_{public}$ = FFHQ. The symbol ↓ (or ↑) indicates smaller (or larger) values are preferred, and the green numbers represent the attack performance improvement. The running time ratio (Ratio) between prior fine-tuning and MI reflects the overhead of fine-tuning.

| Target Model | Method | CelebA | | | | FaceScrub | | | |
|---|---|---|---|---|---|---|---|---|---|
| | | Acc@1↑ | Acc@5↑ | KNN Dist↓ | Ratio↓ | Acc@1↑ | Acc@5↑ | KNN Dist↓ | Ratio↓ |
| **ResNet-18** | PPA | 80.80 | 91.54 | 0.7374 | / | 83.19 | 95.89 | 0.7996 | / |
| | + PPDG-PW (ours) | 83.15 (+2.35) | 94.73 (+3.19) | 0.7082 (-0.0292) | 2.25 | 84.44 | 95.88 | 0.7939 (-0.0057) | 1.70 |
| | + PPDG-CT (ours) | 87.32 (+6.52) | 96.73 (+5.19) | 0.6754 (-0.0620) | 1.57 | 85.70 | 96.53 | 0.7768 (-0.0228) | 1.19 |
| | + PPDG-MMD (ours) | 88.53 (+7.73) | 97.15 (+5.61) | 0.6795 (-0.0579) | 1.13 | 87.02 | 97.13 | 0.7708 (-0.0288) | 0.85 |
| **DenseNet-121** | PPA | 76.74 | 89.04 | 0.7556 | / | 77.13 | 90.47 | 0.7917 | / |
| | + PPDG-PW (ours) | 78.41 (+1.67) | 92.88 (+3.84) | 0.7219 (-0.0337) | 1.67 | 78.45 | 92.89 | 0.7778 (-0.0139) | 1.55 |
| | + PPDG-CT (ours) | 82.51 (+5.77) | 94.81 (+5.77) | 0.7003 (-0.0553) | 1.14 | 84.93 | 96.14 | 0.7405 (-0.0512) | 1.07 |
| | + PPDG-MMD (ours) | 84.02 (+7.28) | 95.37 (+6.33) | 0.6964 (-0.0592) | 0.81 | 85.55 | 96.20 | 0.7363 (-0.0554) | 0.79 |
| **ResNeSt-50** | PPA | 64.52 | 82.79 | 0.8382 | / | 73.65 | 90.96 | 0.8386 | / |
| | + PPDG-PW (ours) | 67.66 (+3.14) | 86.73 (+3.94) | 0.8181 (-0.0201) | 1.68 | 74.98 | 92.24 | 0.8190 (-0.0196) | 1.58 |
| | + PPDG-CT (ours) | 72.57 (+8.05) | 89.66 (+6.87) | 0.7802 (-0.0580) | 1.11 | 77.77 | 93.51 | 0.8045 (-0.0341) | 1.07 |
| | + PPDG-MMD (ours) | 72.99 (+8.47) | 90.01 (+7.22) | 0.7874 (-0.0508) | 0.80 | 78.35 | 93.42 | 0.8109 (-0.0277) | 0.79 |

Table 2: Comparison of MI performance with white-box MIAs in low-resolution settings. Target model M = VGG16 trained on $\mathcal{D}_{private}$ = CelebA. GANs are trained on $\mathcal{D}_{public}$ = CelebA or FFHQ.

| Method | CelebA | | | | FFHQ | | | |
|---|---|---|---|---|---|---|---|---|
| | Acc@1↑ | Acc@5↑ | KNN Dist↓ | Ratio↓ | Acc@1↑ | Acc@5↑ | KNN Dist↓ | Ratio↓ |
| GMI | 17.59 | 39.20 | 1720.46 | / | 8.78 | 23.42 | 1777.72 | / |
| + PPDG-vanilla (ours) | 22.66 (+5.07) | 45.35 (+6.15) | 1697.08 (-23.38) | 2.33 | 9.89 (+1.11) | 25.21 (+1.79) | 1768.62 (-9.10) | 1.62 |
| LOM (GMI) | 64.54 | 86.92 | 1403.81 | / | 38.17 | 64.67 | 1520.77 | / |
| + PPDG-vanilla (ours) | 76.89 (+12.35) | 92.77 (+5.85) | 1329.02 (-74.79) | 1.61 | 50.76 (+12.59) | 77.18 (+12.51) | 1434.71 (-86.06) | 1.70 |
| KEDMI | 72.82 | 93.41 | 1329.12 | / | 41.15 | 70.08 | 1447.15 | / |
| + PPDG-vanilla (ours) | 76.18 (+3.36) | 95.48 (+2.07) | 1307.42 (-21.70) | 28.59 | 41.48 (+0.33) | 73.58 (+3.50) | 1440.48 (-6.67) | 55.20 |
| LOM (KEDMI) | 86.48 | 98.97 | 1249.82 | / | 57.56 | 85.96 | 1384.55 | / |
| + PPDG-vanilla (ours) | 87.21 (+0.73) | 99.00 (+0.03) | 1247.69 (-2.13) | 28.09 | 62.02 (+4.46) | 87.21 (+1.88) | 1353.36 (-31.19) | 56.30 |

tuning captures characteristics of the local data distribution by handling batches of pseudo-private data, whereas point-wise tuning focuses on individual data points. Furthermore, batch-wise tuning is more robust against outliers, leading to a more reliable adjustment of the prior distribution.

**Comparison with white-box MIAs in the low-resolution setting.** For each baseline setup, we report results for PPDG-vanilla. The results are shown in Tab. 2, where PPDG-vanilla consistently outperforms various baseline white-box attacks. The improvement is evident in both attack accuracy and KNN distance metrics. Notably, even with a significant distribution shift between the private training dataset (CelebA) and the public auxiliary dataset (FFHQ), the principled vanilla fine-tuning strategy with original GAN training objectives effectively enhances the density around pseudo-private samples. As a highlight, PPDG-vanilla outperforms the baseline LOM (GMI) by achieving a 12.35% increase in top-1 attack accuracy and reducing KNN distance by approximately 75. Qualitative results of the reconstructed sample are provided in Figs. 11 and 12 in Appx. D.3.

**Attacks against SOTA MI defense methods.** We extend our evaluation to include state-of-the-art model inversion defense methods, specifically BiDO-HSIC and NegLS, comparing the perfor-

Table 3: MI performance against SOTA defense methods in high-resolution settings. The target model M = ResNet-152 is trained on $\mathcal{D}_{private}$ = FaceScrub, GANs are pre-trained on $\mathcal{D}_{public}$ = FFHQ. **Bold** numbers indicate superior results.

| Method | Acc@1↑ | KNN Dist↓ |
|---|---|---|
| No Def. | 77.85 | 0.8235 |
| BiDO-HSIC | 52.50 | 0.9546 |
| + PPDG-PW | 54.65 | 0.9270 |
| + PPDG-CT | 57.40 | 0.9051 |
| + PPDG-MMD | **58.55** | **0.9017** |
| NegLS | 11.35 | 1.3051 |
| + PPDG-PW | 14.65 | 1.2234 |
| + PPDG-CT | **16.25** | 1.2233 |
| + PPDG-MMD | 13.25 | **1.2187** |

mance of our proposed methods—PPDG-PW, PPDG-CT, and PPDG-MMD—with the baseline PPA. As summarized in Tab. 3, each proposed method consistently outperforms the baseline. Notably, PPDG-MMD achieves a 6.05% improvement in top-1 attack accuracy and reduces KNN distance by 0.0529 relative to the baseline against BiDO-HSIC. Similarly, against NegLS, PPDG-CT shows a 4.90% improvement in top-1 attack accuracy and a 0.0818 reduction in KNN distance compared to the baseline. Additional results for the low-resolution setting are provided in Appx. D.1.

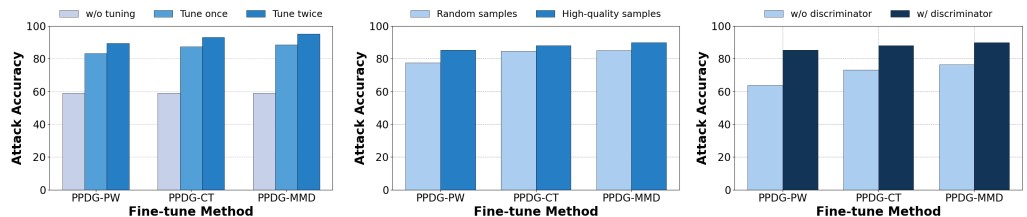

Figure 5: **Ablation study in the high-resolution setting.** Left: Impact of iterative fine-tuning. Middle: Importance of selecting high-quality pseudo-private data for fine-tuning. Right: Effectiveness of using the discriminator as an empirical density estimator to locate high-density neighbors.

### 4.3 Ablation Study

In this section, we present part of the ablation study on MIA in the high-resolution setting to further explore PPDG-MI. The target model is ResNet-18 trained on CelebA; GANs are pre-trained on FFHQ. Additional ablation results for high- and low-resolution settings are provided in Appx. D.2. Discussions (*e.g.,* broader impact, failure case analysis and limitations) are provided in Appx. E.

**Iterative fine-tuning.** The iterative fine-tuning process is crucial for PPDG-MI (*cf.* Algs. 1 and 2). Its goal is to progressively increase the probability of sampling pseudo-private data with closer characteristics to the actual private training data. Ideally, if the classifier has learned all discriminative information of the target identity, this process can continue until it is capable of sampling the actual training data. As shown in left panel of Fig. 5, the attack performance consistently improves with additional rounds of fine-tuning, demonstrating the effectiveness of this approach.

**Selecting high-quality pseudo-private data for density enhancement.** The rationale behind enhancing the density around high-quality pseudo-private data, rather than random reconstructed ones, is that the former better reflect the characteristics of the private training data and are semantically closer. Thus, this increases the probability of sampling the actual training data. The middle panel of Fig. 5 compares the attack results of enhancing density around high-quality pseudo-private samples and randomly selected recovered samples, demonstrating the effectiveness of this strategy.

**Locating high-density neighbors using the discriminator.** We investigate the effect of using the discriminator D as an empirical density estimator to locate samples in high-density areas in Eqs. (6), (9a), and (10a). The comparison results, with and without the discriminator, are shown in the right panel of Fig. 5. The results indicate that MI performance decreases significantly without using the discriminator, with an approximate 13-22% reduction in attack accuracy across different fine-tuning methods. This demonstrates the effectiveness of incorporating D as a density estimator.

## 5 Conclusion

In this paper, we identify a fundamental limitation common to state-of-the-art generative MIAs, *i.e.,* the utilization of a *fixed prior* during the inversion phase. We argue that this approach is sub-optimal for MIAs. Accordingly, we introduce a novel inversion pipeline called *pseudo-private data guided* MI (PPDG-MI), which, for the first time, involves iteratively tuning the distributional prior during the inversion process using pseudo-private samples. This increases the probability of recovering actual private training data. We propose multiple realizations of PPDG-MI and demonstrate their effectiveness through extensive experiments. Our findings pave the way for future research on generative MIAs and highlight the urgent need for more robust defenses against MIAs.

## Acknowledgments

XP and BH were supported by NSFC General Program No. 62376235, Guangdong Basic and Applied Basic Research Foundation Nos. 2022A1515011652 and 2024A1515012399, HKBU Faculty Niche Research Areas No. RC-FNRA-IG/22-23/SCI/04, and HKBU CSD Departmental Incentive Scheme. TLL was partially supported by the following Australian Research Council projects: FT220100318, DP220102121, LP220100527, LP220200949, and IC190100031. FL is supported by the Australian Research Council (ARC) with grant numbers DP230101540 and DE240101089, and the NSF&CSIRO Responsible AI program with grant number 2303037.

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

# Appendix

# A   Detailed Related Work and Preliminary

## A.1   Model Inversion Attacks

*Model inversion attacks* (MIAs) were first introduced by Fredrikson et al. [2014] as a method to reconstruct private data from outputs in simple regression tasks with shallow models. This groundbreaking research highlighted the privacy risks of exposing sensitive data via model predictions. Based on this foundation, Zhang et al. [2020] extended MIAs to more complex DNNs. They developed a methodology that involved learning a distributional prior from a publicly available auxiliary dataset, allowing for effective MIAs in a constrained latent space of the generator. Since these foundational studies, MIAs have received increased attention, particularly in the realm of high-dimensional image data. Recent research divides MIAs into three types based on the attacker's access to the model: white-box, black-box, and label-only settings. Each category represents varying levels of accessibility and potential risk, which informs the ongoing development of defensive strategies in this area.

In the white-box setting, attackers have full access to the model, including its architecture and weights. The first white-box attack on DNNs, *generative model inversion* attack [Zhang et al., 2020], utilized *generative adversarial networks* (GANs) to learn a distributional prior and optimize within the latent space. Subsequently, the *knowledge-enriched distributional model inversion* (KEDMI) attack [Chen et al., 2021] employed a specialized GAN with an advanced discriminator that leverages information from the target model. Wang et al. [2021a] proposed *variational model inversion* (VMI), which uses a probabilistic approach with a variational objective to ensure both diversity and accuracy. This evolution culminated in the *Plug & Play Attack* (PPA) [Struppek et al., 2022], which enhances the recovery of images ranging from low to high resolution. Additionally, Nguyen et al. [2023] introduced *logit maximization* (LOM) loss as an alternative to the *cross-entropy* (CE) identity loss previously used in [Zhang et al., 2020, Chen et al., 2021, Wang et al., 2021a] and addressed issues of model overfitting with the model augmentation technique. Moreover, Yuan et al. [2023] advanced MIAs with the *pseudo label-guided model inversion* (PLG-MI), employing a *conditional* GAN (cGAN) and max-margin loss, and using pseudo-labels to decouple the search space for different classes.

In the black-box setting, attackers lack direct access to the model's internals but can still query the target model a predetermined number of times to observe outputs for specific inputs, using this data to infer sensitive information indirectly. An et al. [2022] adopted a genetic search algorithm as an alternative to gradient descent in the black-box setting. While Han et al. [2023] developed the *reinforcement learning-based model inversion* (RLB-MI) algorithm, formulating the latent vector optimization as a *Markov decision process* (MDP) using reinforcement learning.

The label-only setting poses the greatest challenge in model inversion attacks, where attackers only have access to the hard labels produced by the model, without any confidence scores or other related information. Kahla et al. [2022] tackled this issue using the *boundary-repelling model inversion* (BREP-MI) algorithm, which effectively utilizes the labels to simulate a gradient through max-margin loss. This method effectively navigates toward the concentrated areas of the target class in the data distribution by estimating model predictions across a conceptual sphere. Inspired by transfer learning (TL), Nguyen et al. [2024] proposed the *label-only via knowledge transfer* (LOKT) method for label-only model inversion, transferring knowledge from the target model to a *target model-assisted* ACGAN (T-ACGAN), effectively turning the label-only scenario into a white-box setting.

## A.2   Distribution Discrepancy Measure

This section introduces two measures used to evaluate the closeness between distributions P and Q: *maximum mean discrepancy* (MMD) and *conditional transport* (CT).

**Maximum Mean Discrepancy (MMD).** Given two random variables $X \sim P$, $Y \sim Q$, the MMD measure is defined as follows:

$$\mathrm{MMD}(P, Q; \mathcal{F}) := \sup_{f \in \mathcal{F}} |\mathbb{E}[f(X)] - \mathbb{E}[f(Y)]|, \tag{11}$$

where $\mathcal{F}$ is a class of functions [Gretton et al., 2012a]. This class is often restricted to a unit ball in a *reproducing kernel Hilbert space* (RKHS) to facilitate analytical solutions [Gretton et al., 2012a],

leading to the kernel-based MMD defined in the following,

$$\text{MMD}(\text{P}, \text{Q}; \mathcal{H}_k) := \sup_{f \in \mathcal{H}, \|f\|_{\mathcal{H}_k} \leq 1} |\mathbb{E}[f(X)] - \mathbb{E}[f(Y)]|, \tag{12}$$

where $k$ is a bounded kernel chosen based on the specific properties of the RKHS $\mathcal{H}_k$. The use of RKHS allows for the effective computation of MMD, leveraging kernel functions to measure the distance between the distributions in a high-dimensional feature space.

**Estimation of MMD.** Given sample sets $S_X = \{\mathbf{x}_i\}_{i=1}^n \sim \text{P}$ and $S_Y = \{\mathbf{y}_j\}_{j=1}^m \sim \text{Q}$, MMD (Eq. (12)) can be estimated with the $U$-statistic estimator, which is unbiased for $\text{MMD}^2$ [Gretton et al., 2012a]:

$$\widehat{\text{MMD}}_u^2(S_X, S_Y; k) = \frac{1}{n(n-1)} \sum_{i=1}^n \sum_{j=1, j \neq i}^n k(x_i, x_j) + \frac{1}{m(m-1)} \sum_{i=1}^m \sum_{j=1, j \neq i}^m k(y_i, y_j)$$

$$- \frac{2}{mn} \sum_{i=1}^n \sum_{j=1}^m k(x_i, y_j). \tag{13}$$

where $\mathbf{x}_i, \mathbf{x}_j \in S_X$ and $\mathbf{y}_i, \mathbf{y}_j \in S_Y$. This estimator efficiently computes the MMD by aggregating kernel evaluations over pairs of samples from both distributions.

**Conditional Transport (CT).** The CT measure provides a complementary approach to MMD by focusing on the transport cost between distributions. It consists of two components:

$$\text{CT}(\text{P}, \text{Q}) := \mathcal{L}_{X \to Y} + \mathcal{L}_{Y \to X}, \tag{14}$$

where $\mathcal{L}_{X \to Y}$ and $\mathcal{L}_{Y \to X}$ represent the transport costs for the forward and backward CT, respectively. They are defined as follows:

$$\mathcal{L}_{X \to Y} := \mathbb{E}_{\mathbf{x} \sim \text{P}(X)} \mathbb{E}_{\mathbf{y} \sim \Pi(\cdot \,|\, \mathbf{x})} [c(\mathbf{x}, \mathbf{y})], \tag{15a}$$

$$\mathcal{L}_{Y \to X} := \mathbb{E}_{\mathbf{y} \sim \text{Q}(Y)} \mathbb{E}_{\mathbf{x} \sim \Pi(\cdot \,|\, \mathbf{y})} [c(\mathbf{x}, \mathbf{y})], \tag{15b}$$

where $\Pi(Y \,|\, X) = \frac{e^{-d_\psi(X,Y)} Q(Y)}{\int e^{-d_\psi(X,Y)} Q(Y) \mathrm{d}Y}$, and $\Pi(X \,|\, Y) = \frac{e^{-d_\psi(X,Y)} P(X)}{\int e^{-d_\psi(X,Y)} P(X) \mathrm{d}X}$ represent the conditional distributions of $Y$ given $X$ and $X$ given $Y$, respectively. Here, $d_\psi(X, Y)$ is a function parameterized by $\psi$ that measures the distance between $X$ and $Y$, and $c(\mathbf{x}, \mathbf{y})$ is a cost function that measures the distance between the points $\mathbf{x}$ and $\mathbf{y}$.

**Estimation of CT.** Given sample sets $S_X = \{\mathbf{x}_i\}_{i=1}^n \sim \text{P}$ and $S_Y = \{\mathbf{y}_j\}_{j=1}^m \sim \text{Q}$, the CT measure can be approximated as follows:

$$\begin{aligned}
\text{CT}(S_X, S_Y) &= \mathcal{L}_{X \to \hat{Y}} + \mathcal{L}_{Y \to \hat{X}} \\
&= \mathbb{E}_{\mathbf{y}_{1:m} \overset{iid}{\sim} Q(Y)} \mathbb{E}_{\mathbf{x} \sim \text{P}(X)} \left[ \sum_{j=1}^m c(\mathbf{x}, \mathbf{y}_j) \hat{\Pi}(\mathbf{y}_j \,|\, \mathbf{x}) \right] \\
&+ \mathbb{E}_{\mathbf{x}_{1:n} \overset{iid}{\sim} P(X)} \mathbb{E}_{\mathbf{y} \sim \text{Q}(Y)} \left[ \sum_{i=1}^n c(\mathbf{x}_i, \mathbf{y}) \hat{\Pi}(\mathbf{x}_i \,|\, \mathbf{y}) \right] \\
&= \sum_{i=1}^n \sum_{j=1}^m c(\mathbf{x}_i, \mathbf{y}_j) \left( \frac{e^{-d_\psi(\mathbf{x}_i, \mathbf{y}_j)}}{\sum_{j'=1}^m e^{-d_\psi(\mathbf{x}_i, \mathbf{y}_{j'})}} + \frac{e^{-d_\psi(\mathbf{x}_i, \mathbf{y}_j)}}{\sum_{i'=1}^n e^{-d_\psi(\mathbf{x}_{i'}, \mathbf{y}_j)}} \right). \tag{16}
\end{aligned}$$

The CT measure evaluates the cost of transporting samples from one distribution to another, providing a detailed assessment of how closely the distributions align.

---

**Algorithm 1** Pseudo-Private Data Guided Model Inversion with Vanilla Tuning.

---

**Input:** Target model M, pre-trained generator $G(\cdot; \boldsymbol{\theta})$, pre-trained discriminator $D(\cdot; \boldsymbol{\phi})$, public auxiliary dataset $\mathcal{D}_{\text{public}}$, number of fine-tuning rounds $R$, and the set of identities to be reconstructed $C$.

---

1: $\boldsymbol{\theta}_{old} \leftarrow \boldsymbol{\theta}$;
2: `reconstructed_samples = [];`
3: **for** each target identity $y$ in $C$ **do**
4:    $\boldsymbol{\theta} \leftarrow \boldsymbol{\theta}_{old}$;
5:    **for** round $= 1, \ldots, R$ **do**
6:       # Step-1.  Model inversion on target model M
7:       **Initialize** latent codes: $\mathbf{Z} = \{\mathbf{z}_i \mid \mathbf{z}_i \in \mathcal{Z}, i = 1, \ldots, N\}$;
8:       **Obtain** optimized latent codes $\hat{\mathbf{Z}}$ using Eq. (1);
9:       # Step-2.  Select high-quality pseudo-private samples
10:      **Select** pseudo-private samples with top-$K$ stable prediction scores: $\mathcal{D}_{\text{private}}^{\text{s}'} = $ $\text{TopK}\{\mathbb{E}[\mathbb{P}_M(y \mid T(\hat{\mathbf{x}}))] \mid \hat{\mathbf{x}} \in \mathcal{D}_{\text{private}}^{\text{s}}\}$;
11:      # Step-3.  Enhance density around $\mathcal{D}_{\text{private}}^{\text{s}'}$

        **Fine-tune** $G(\cdot; \boldsymbol{\theta})$ and $D(\cdot; \boldsymbol{\phi})$ to enhance density around $\mathcal{D}_{\text{private}}^{\text{s}'}$ by directly fine-tuning them with the original GAN training objective on $\mathcal{D}_{\text{public}} \cup \mathcal{D}_{\text{private}}^{\text{s}'}$;
12:    **end for**
13:    `reconstructed_samples +=` $\mathcal{D}_{\text{private}}^{\text{s}}$;
14: **end for**
15: **Output:** `reconstructed_samples.`

---

**Algorithm 2** Pseudo-Private Data Guided Model Inversion with Point-wise or Batch-wise Tuning.

---

**Input:** Target model M, pre-trained generator $G(\cdot; \boldsymbol{\theta})$, pre-trained discriminator D, number of fine-tuning rounds $R$, identity set to be reconstructed $C$; point-wise tuning flag `PW_Flag`, distribution discrepancy measure $\delta$.

---

1: $\boldsymbol{\theta}_{old} \leftarrow \boldsymbol{\theta}$;
2: `reconstructed_samples = [];`
3: **for** each target identity $y$ in $C$ **do**
4:    $\boldsymbol{\theta} \leftarrow \boldsymbol{\theta}_{old}$;
5:    **for** round $= 1, \ldots, R$ **do**
6:       # Step-1.  Model inversion on target model M
7:       **Initialize** latent codes: $\mathbf{Z} = \{\mathbf{z}_i \mid \mathbf{z}_i \in \mathcal{Z}, i = 1, \ldots, N\}$;
8:       **Obtain** optimized latent codes $\hat{\mathbf{Z}}$ using Eq. (1);
9:       **Generate** pseudo-private dataset: $\mathcal{D}_{\text{private}}^{\text{s}} = \{\hat{\mathbf{x}} = G(\hat{\mathbf{z}}) \mid \hat{\mathbf{z}} \in \hat{\mathbf{Z}}\}$;
10:      # Step-2.  Select high-quality pseudo-private samples
11:      **Select** pseudo-private samples with top-$K$ stable prediction scores: $\mathcal{D}_{\text{private}}^{\text{s}'} = $ $\text{TopK}\{\mathbb{E}[\mathbb{P}_M(y \mid T(\hat{\mathbf{x}}))] \mid \hat{\mathbf{x}} \in \mathcal{D}_{\text{private}}^{\text{s}}\}$;
12:      # Step-3.  Enhance density around $\mathcal{D}_{\text{private}}^{\text{s}'}$
13:    **if** `PW_Flag` **then**
14:      # Point-wise tuning
15:      **Locate** high-density neighbors $\mathbf{Z}^{\text{p}}$ by optimizing Eq. (6);
16:      **Fine-tune** $G(\cdot; \boldsymbol{\theta})$ to enhance density around $\mathcal{D}_{\text{private}}^{\text{s}'}$ by optimizing Eq. (7);
17:    **else**
18:      # Batch-wise tuning
19:      **if** $\delta$ is `MMD` **then**
20:        **Locate** high-density neighbors $\mathbf{Z}^{\text{p}}$ by optimizing Eq. (9a);
21:        **Fine-tune** $G(\cdot; \boldsymbol{\theta})$ to enhance density around $\mathcal{D}_{\text{private}}^{\text{s}'}$ by optimizing Eq. (9b);
22:      **else if** $\delta$ is `CT` **then**
23:        **Locate** high-density neighbors $\mathbf{Z}^{\text{p}}$ by optimizing Eq. (10a);
24:        **Fine-tune** $G(\cdot; \boldsymbol{\theta})$ to enhance density around $\mathcal{D}_{\text{private}}^{\text{s}'}$ by optimizing Eq. (10b);
25:      **end if**
26:    **end if**
27:    **end for**
28:    `reconstructed_samples +=` $\mathcal{D}_{\text{private}}^{\text{s}}$;
29: **end for**
30: **Output:** `reconstructed_samples.`

# B The Algorithmic Realizations of PPDG-MI

This section presents the detailed algorithmic realization of the pseudo-private data guided model inversion (PPDG-MI) method. We describe two variants of the PPDG-MI method, each tailored for different MI scenarios. The first variant utilizes vanilla tuning (*cf.* Alg. 1), applicable for low-resolution MIAs where the adversary trains a GAN from scratch. The second variant employs nuanced point-wise or batch-wise tuning (*cf.* Alg. 2), suitable for high-resolution MIAs (*i.e.,* PPA) where pre-trained generators are provided without access to the original training details. These methods are designed to enhance the density of pseudo-private samples under the prior distribution, thereby increasing the probability of sampling from the private data distribution.

# C Experimental Details

Table 4: A summary of experimental setups.

| Type | MIAs | Private Dataset | Public Dataset | Target Model | Evaluation Model |
|---|---|---|---|---|---|
| White-box | GMI / KEDMI / LOM | CelebA | CelebA / FFHQ | VGG16 / face.evoLVe | face.evoLVe |
| | PPA | CelebA / FaceScrub | CelebA | ResNet-18 / DenseNet-121 / ResNeSt-50 | Inception-v3 |
| Black-box | RLB-MI | CelebA | CelebA | VGG16 | face.evoLVe |
| Label-only | BREP-MI | CelebA | CelebA | VGG16 | face.evoLVe |

## C.1 Hard- and Software Details

In our experiments with *Plug & Play Attacks* (PPA), we conducted all of them on Oracle Linux Server 8.9 using NVIDIA Ampere A100-80G GPUs. The hardware operated under CUDA 11.7, Python 3.9.18, and PyTorch 1.13.1. For MIAs targeting low-resolution facial recognition tasks, we executed these experiments on Ubuntu 20.04.4 LTS, equipped with NVIDIA GeForce RTX 3090 GPUs. This setup utilized CUDA 11.6, Python 3.7.12, and PyTorch 1.13.1.

## C.2 Evaluation Models

For experiments based on PPA, we train Inception-v3 evaluation models, following the code and guidelines available at the repository `https://github.com/LukasStruppek/Plug-and-Play-Attacks`. For training details, please refer to [Struppek et al., 2022]. These models achieve test accuracies of $96.53\%$ on FaceScrub and $94.87\%$ on CelebA. In addition, We use the pre-trained FaceNet [Schroff et al., 2015] from `https://github.com/timesler/facenet-pytorch` to compute the K-nearest neighbors distance, providing a measure of similarity between training samples and the reconstructed samples on the facial recognition tasks.

In experiments involving classifiers trained on the $64 \times 64$ resolution CelebA dataset, we utilize an evaluation model available for download at repository `https://github.com/sutd-visual-computing-group/Re-thinking_MI`. This model is built upon the face.evoLVe model [Wang et al., 2021b], incorporating a modified ResNet50 backbone, and achieves a stated test accuracy of $95.88\%$. For training details, please refer to Zhang et al. [2020].

## C.3 Target Models

For training target models on $224 \times 224$ resolution CelebA and FaceScrub images, we adapt the training scripts and hyperparameters provided in the corresponding code repository and described in [Struppek et al., 2022]. The only training parameter we modify is the smoothing factor of the label smoothing loss. All models are trained for 100 epochs using the Adam optimizer [Kingma and Ba, 2015], with an initial learning rate of $10^{-3}$ and $\beta = (0.9, 0.999)$, and a weight decay of $10^{-3}$. We reduce the learning rate by a factor of 0.1 after 75 and 90 epochs. The batch size is set to 128. All

data samples are normalized with $\mu = \sigma = 0.5$ and resized to $224 \times 224$. The training samples are then augmented by random cropping with a scale of $[0.85, 1.0]$ and a fixed ratio of $1.0$. Crops are resized back to $224 \times 224$, and samples are horizontally flipped in $50\%$ of the cases.

For training target models on $64 \times 64$ resolution CelebA images, we use the training script provided at https://github.com/sutd-visual-computing-group/Re-thinking_MI. These models are trained for 50 epochs using the SGD optimizer with an initial learning rate of $10^{-2}$, a momentum term of $0.9$, and a weight decay of $10^{-4}$. The batch size is set to $64$. The learning rate decay schedule varies depending on the model; please refer to the training script for details.

### C.4 Attack Parameters

PPA consists of three stages: pre-attack latent code selection, MIA optimization, and final results selection. During pre-attack latent code selection, we choose 100 candidates for each target identity from a search space of 500 latent codes for both CelebA and FaceScrub. In the MIA optimization phase, we maintain an equal number of queries to the target model $\mathbf{M}$ to ensure a fair comparison for both the baseline attack and PPDG-MI. Thus, samples are optimized for 70 steps for both CelebA and FaceScrub in the baseline attack and 35 steps for each round of MIA in PPDG-MI. The final results selection stage is omitted. We target the first 100 identities and generate 100 samples per identity. Nonetheless, after some consideration, we believe that maintaining an equal number of queries to the target model $\mathbf{M}$ in each round of MIA for both PPDG-MI and the baseline attack may be a more reasonable approach to evaluate the performance of the proposed method.

For low-resolution attacks, we generate $1,000$ samples per identity for CelebA and $2,000$ samples per identity for FFHQ, as training GANs typically require more samples. Specifically, for low-resolution white-box attacks, we target the first $100$ identities for CelebA and the first $50$ identities for FFHQ. Samples in GMI and LOM (GMI) are optimized for $2,400$ steps on the VGG16 target model and $1,200$ steps on face.evoLVe, while KEDMI and LOM (KEDMI) optimize samples for $1,200$ steps on both target models. For PLG-MI, we target $50$ identities of the CelebA dataset, generating $200$ samples per identity. Samples are optimized for $80$ iterations on VGG16 and face.evoLVe target models. For black-box and label-only attacks, we limit our selection to the first $10$ identities due to the extremely time-consuming nature of point-wise optimization in these settings. In the black-box attack (*i.e.,* RLB-MI), samples undergo $10,000$ optimization steps in the baseline attack and in each round of PPDG-MI. For the label-only attack (BREP-MI), the maximum number of optimization steps is set to $1,000$ for both the baseline and each round of PPDG-MI.

Due to the significant time required for MIAs, we perform a single attack against each target model. To reduce randomness, we generate at least 100 samples for each target class across various setups.

### C.5 Evaluation Metrics

**Attack Accuracy (Attack Acc).** Following Zhang et al. [2020], we use an *evaluation model* (typically more powerful than the target model) trained on the target model's training data to identify reconstructed images (*cf.* Tab. 4). Intuitively, it can be viewed as a proxy for a human evaluator. Attack accuracy is calculated as the proportion of predictions matching the target identity; top-1 (Acc@1) and top-5 (Acc@5) attack accuracy is adopted.

**K-Nearest Neighbors Distance (KNN Dist).** KNN Dist represents the $l_2$ distance between reconstructed images and the nearest samples from the target model's training data in the embedding space, indicating visual similarity between faces. For PPA [Struppek et al., 2022], we use the penultimate layer of pre-trained FaceNet [Schroff et al., 2015], whereas for other low-resolution MIAs, we adopt the penultimate layer of the evaluation model. Lower distances indicate a closer resemblance between the reconstructed samples and the training data.

### C.6 Experimental Details for Fig. 1

We present the experimental details for generating Fig. 1. In the motivation-driven experiments, we evaluate the distribution discrepancy between commonly adopted private training datasets and public auxiliary datasets, and investigate the impact of distribution discrepancy on attack performance of MIAs using various public auxiliary datasets and target models.

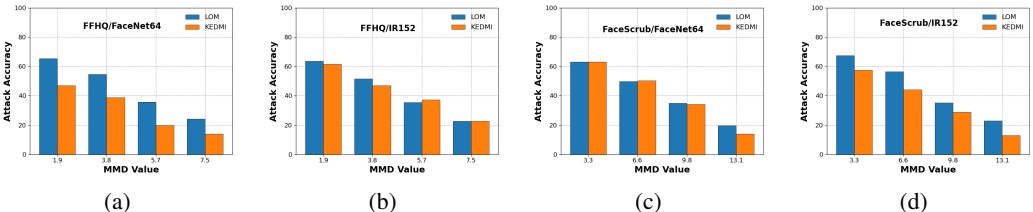

(a)          (b)          (c)          (d)

Figure 6: **Impact of distribution discrepancies on MIAs across various settings.** The attack performance of MIAs is analyzed under four distinct combinations of public auxiliary datasets $\mathcal{D}_{\text{public}}$ and target models M, with the same private training dataset $\mathcal{D}_{\text{private}}$ = CelebA: (a) $\mathcal{D}_{\text{public}}$ = FFHQ and M = face.evoLVe, (b) $\mathcal{D}_{\text{public}}$ = FFHQ and M = IR152, (c) $\mathcal{D}_{\text{public}}$ = FaceScrub and M = face.evoLVe, and (d) $\mathcal{D}_{\text{public}}$ = FaceScrub and M = IR152. The attack performance consistently diminishes as the discrepancy between the $\mathcal{D}_{\text{private}}$ (CelebA) and $\mathcal{D}'_{\text{public}}$ increases.

In Fig. 1(a), we employ kernel-based two-sample tests to evaluate the distributional discrepancy between public auxiliary datasets (FFHQ and FaceScrub) and the private dataset (CelebA). A p-value of 0 signifies no statistical basis to reject the null hypothesis, indicating no discernible distribution difference between the datasets. Conversely, a p-value of 1 implies definitive evidence to reject the null hypothesis. We utilize a Gaussian Kernel-based test to calculate p-values at the feature level, extracting a 512-dimensional feature vector from each image using the face.evoLVe feature extractor. We analyze subsets of N feature vectors, with N varying from $1,000$ to $10,000$. Each subset is sampled 20 times from both datasets, and the results represent the average of these samples.

In Figs. 1(b) and 1(c), we investigate the impact of distribution discrepancy on the attack performance of MIAs (LOM and KEDMI) using VGG16 as the target model and face.evoLVe as the evaluation model. The private dataset is CelebA, and the public auxiliary datasets are FFHQ and FaceScrub. We construct the proxy public auxiliary dataset by incrementally integrating the private data into the public auxiliary dataset, increasing the private data ratio by $20\%$ at each interval, from $20\%$ to $80\%$. To manage computational demands, we measure MMD across batches of 250 images each, using Gaussian kernels with a bandwidth of 1024. The final MMD value is the average result across all batches. Additional results involving IR152 and face.evoLVe target models are shown in Fig. 6.

## C.7 Experimental Details for Toy Example From Sec. 3.3

The target model used in the toy example is a simple 3-layer *multilayer perceptron* (MLP). This MLP comprises two hidden layers, with the first containing 100 neurons and the second 200 neurons. Each hidden layer is followed by a *rectified linear unit* (ReLU) activation function to introduce non-linearity. The model is trained on a dataset sampled from a 3-class class-conditional Gaussian distribution, employing standard cross-entropy for loss calculation. Training is performed for $6,000$ epochs using standard *stochastic gradient descent* (SGD) with an initial learning rate of $0.5$, enhanced by a linear warm-up schedule to increase the learning rate gradually.

The public auxiliary dataset, distinct from the training dataset to prevent distribution overlap, is generated from a separate Gaussian distribution. The distributional prior is learned using a GAN, where both the generator and discriminator are structured as MLPs with three hidden layers of 400 neurons each. All layers in both models are followed by ReLU activations. The GAN is trained under the *Wasserstein GAN with gradient penalty* (WGAN-GP) [Gulrajani et al., 2017] framework to ensure a stable training process and reliable generation of new data samples.

We employ the learned prior to guide the attack targeting Class 1. Initially, we randomly generate $1,000$ initial points and iteratively update them to maximize the model's prediction score for Class 1 through minimizing an identity loss (*i.e.,* cross-entropy loss), following the approach proposed by Zhang et al. [2020]. We optimize this process using SGD with a learning rate of $0.1$.

To ensure a fair comparison between the baseline and PPDG-MI, we ensure that both attacks make the same number of queries to the model. Specifically, we query the model $1,000$ times in the baseline attack. For PPDG-MI, we first conduct one round of model inversion, making 500 queries to the target model to generate $1,000$ pseudo-private data points. Subsequently, we enhance the density of the pseudo-private data under the prior distribution by directly aligning the distribution between the

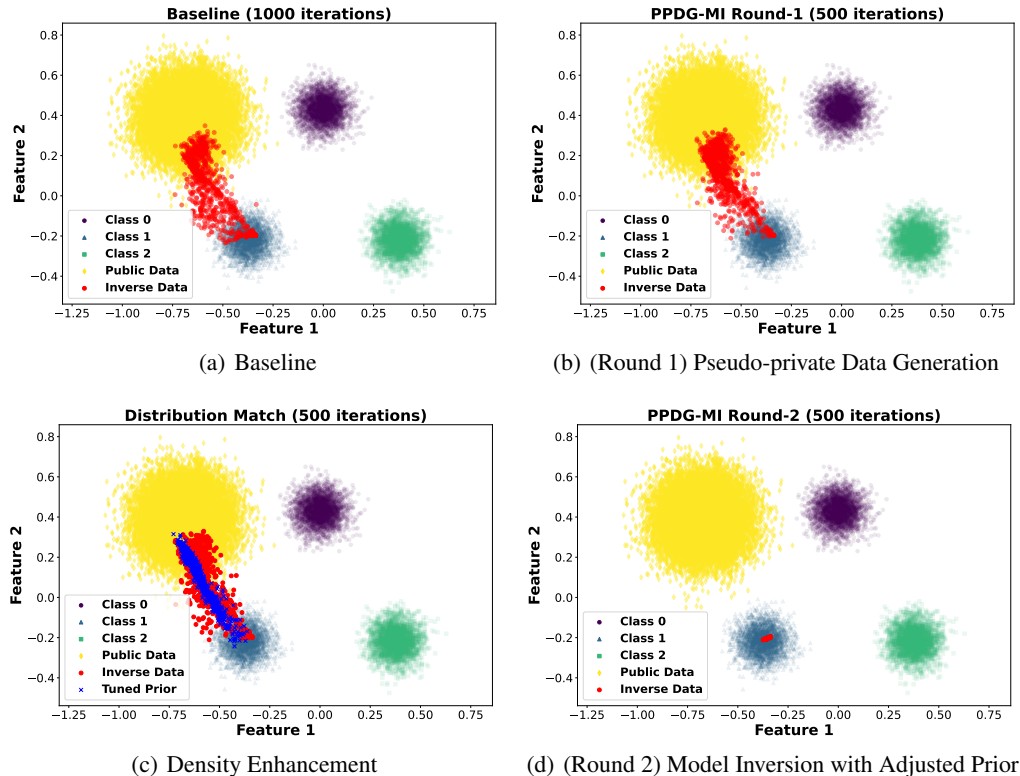

| (a) Baseline | (b) (Round 1) Pseudo-private Data Generation |
| :---: | :---: |

| (c) Density Enhancement | (d) (Round 2) Model Inversion with Adjusted Prior |
| :---: | :---: |

Figure 7: **Illustration of the rationale behind PPDG-MI using a simple 2D example (larger version).** Training samples from Class 0-2 are represented by purple circles, blue triangles, and green squares, respectively, while public auxiliary data are depicted as yellow diamonds. MIAs aim to recover training samples from Class 1. Reconstructed samples by MIAs are shown as red circles. (a) Attack results of the baseline attack with a fixed prior. (b) Pseudo-private data generation. (c) Enhancing the density of pseudo-private data under prior distribution. (d) The final attack results of PPDG-MI with the tuned prior, where all the recovered points converge to the centroid of the class distribution, indicating the most representative features are revealed.

prior distribution and the empirical pseudo-private data distribution by minimizing the distribution discrepancy between them. Then, we apply the fine-tuned prior to guide the second round of model inversion, making 500 queries to the model.

The baseline attack results are shown in Fig. 7(a) and the final attack results of PPDG-MI are shown in Fig. 7(d). It is evident that, in comparison to the baseline where only a small fraction of the reconstructed samples fall within the high-density regions of the training data distribution, all reconstructed samples from PPDG-MI are located in these high-density regions. Quantitatively, the attack performance is evaluated by measuring two metrics: the average distance between the reconstructed samples and the mean of the target class distribution, and the proportion of reconstructed samples that lie in three standard deviations ($3\sigma$) of the mean. In comparison, the baseline achieves an average distance of $0.34$ and attack accuracy of $22.60\%$, and PPDG-MI achieves an average distance of $0.04$ and attack accuracy of $100.00\%$.

### C.8 Investigate Distribution Alignment on High-dimensional Image Data

In this experiment, we aim to extend the density enhancement strategy from the toy experiment to high-dimensional image data. We use the pre-trained StyleGAN on FFHQ as the distributional prior, with ResNet-18 trained on the CelebA dataset as the target model. We generate the pseudo-private dataset $\mathcal{D}^{\text{s}}_{\text{private}}$ by

Table 5: Enhance density of pseudo-private data under the prior distribution by distribution alignment.

| Method | Acc@1↑ | KNN Dist↓ |
| :--- | :---: | :---: |
| PPA | 84.30 | 0.7136 |
| + PPDG with distribution alignment | 17.62 | 1.1718 |

Table 6: Comparison of MI performance with representative white-box MIAs in the low-resolution setting. The target model $\mathrm{M}$ is face.evoLVe trained on $\mathcal{D}_{\text{private}}$ = CelebA. GANs are trained on $\mathcal{D}_{\text{public}}$ = CelebA or FFHQ. The symbol ↓ (or ↑) indicates smaller (or larger) values are preferred, and the green numbers represent the attack performance improvement. The running time ratio (Ratio) between prior fine-tuning and MI reflects the relative overhead of fine-tuning.

| Method | CelebA | | | | FFHQ | | | |
| --- | --- | --- | --- | --- | --- | --- | --- | --- |
| | Acc@1↑ | Acc@5↑ | KNN Dist↓ | Ratio↓ | Acc@1↑ | Acc@5↑ | KNN Dist↓ | Ratio↓ |
| GMI | 26.86 | 50.96 | 1646.93 | / | 11.77 | 28.96 | 1748.30 | / |
| + PPDG-vanilla (ours) | 28.29 (+1.43) | 51.83 (+0.87) | 1635.62 (-11.31) | 0.99 | 12.37 (+0.60) | 30.04 (+1.08) | 1739.78 (-8.52) | 0.80 |
| LOM (GMI) | 67.96 | 87.26 | 1412.14 | / | 38.66 | 65.78 | 1539.27 | / |
| + PPDG-vanilla (ours) | 72.35 (+4.39) | 89.15 (+1.89) | 1378.67 (-33.47) | 1.02 | 48.32 (+9.66) | 73.31 (+7.53) | 1489.34 (-49.93) | 1.03 |
| KEDMI | 87.16 | 98.14 | 1230.81 | / | 55.99 | 82.23 | 1406.98 | / |
| + PPDG-vanilla (ours) | 87.82 (+0.66) | 98.19 (+0.05) | 1225.32 (-5.49) | 17.65 | 57.54 (+1.55) | 83.57 (+1.34) | 1397.32 (-9.66) | 66.44 |
| LOM (KEDMI) | 91.82 | 99.33 | 1275.39 | / | 71.81 | 94.00 | 1379.06 | / |
| + PPDG-vanilla (ours) | 92.11 (+0.29) | 98.52 (-0.81) | 1249.85 (-25.54) | 15.39 | 60.54 (-11.27) | 80.04 (-13.96) | 1441.34 (+62.28) | 64.50 |

Table 7: Comparison of MI performance with PLG-MI in the low-resolution setting. Target model $\mathrm{M}$ = VGG16 or face.evoLVe trained on $\mathcal{D}_{\text{private}}$ = CelebA. GANs are trained on $\mathcal{D}_{\text{public}}$ = FaceScrub.

| Method | VGG16 | | | | face.evoLVe | | | |
| --- | --- | --- | --- | --- | --- | --- | --- | --- |
| | Acc@1↑ | Acc@5↑ | KNN Dist↓ | Ratio↓ | Acc@1↑ | Acc@5↑ | KNN Dist↓ | Ratio↓ |
| PLG-MI | 33.59 | 56.54 | 1496.94 | / | 53.83 | 83.35 | 1430.78 | / |
| + PPDG-vanilla (ours) | 34.70 (+1.11) | 59.32 (+2.78) | 1487.07 (-9.87) | 3.26 | 56.08 (+2.25) | 84.15 (+0.80) | 1405.68 (-25.10) | 2.74 |

conducting MIA on target identities 1-100, recovering 100 samples per identity, resulting in a total of 10,000 samples in $\mathcal{D}_{\text{private}}^{\text{s}}$. We then increase the density around $\mathcal{D}_{\text{private}}^{\text{s}}$ under the prior distribution $\mathrm{P}(\mathcal{X}_{\text{prior}})$ by fine-tuning $\mathrm{G}(\cdot; \boldsymbol{\theta})$ to align with the empirical distribution of $\mathcal{D}_{\text{private}}^{\text{s}}$, using the CT measure. The results are shown in Tab. 5, where we observe a dramatic decrease in MI performance after fine-tuning the generator. This indicates that direct distribution alignment is less effective for higher-dimensional image data, as it disrupts the generator's manifold. Therefore, we need to employ a nuanced tuning strategy with smaller perturbations to the image manifold.

# D    Additional Experimental Results

## D.1    Additional Main Results

**Comparison with white-box MIAs in the low-resolution setting.** In this experiment, we utilize the face.evoLVe as the target model. The results are presented in Tab. 6, where PPDG-vanilla consistently outperforms various baseline white-box attacks, offering notable improvements in both attack accuracy and KNN distance metrics across two public auxiliary datasets, CelebA and FFHQ. For instance, in the LOM (GMI) setup, adding PPDG-vanilla results in an increase in top-1 attack accuracy from $68.09\%$ to $71.39\%$ for CelebA, and top-5 attack accuracy from $87.31\%$ to $88.12\%$. Additionally, there is a reduction in KNN distance from $1417.23$ to $1385.10$ for CelebA. Similarly, significant improvements are observed for FFHQ, where attack accuracy increases and KNN distances decrease, indicating enhanced data density around pseudo-private samples. However, there is a failure case involving the setup where the target model is face.evoLVe trained on CelebA, with the public dataset as FFHQ. We attribute this failure to multiple factors, which are analyzed in Appx. E. Generally, these results suggest that even with substantial distribution shifts between the private dataset CelebA and the public dataset FFHQ, our principled vanilla fine-tuning strategy, which retains the original GAN training objectives, can effectively improve MIA performance.

**Comparison with PLG-MI in the low-resolution setting.** In this setting, we use the state-of-the-art white-box attack PLG-MI [Yuan et al., 2023] as the baseline for comparison. PLG-MI leverages the target model to generate pseudo-labels for public auxiliary datasets, enabling the training of a cGAN (conditional GAN) on this labeled data. This further decouples the search space across different classes. However, the poor visual quality of samples generated by PLG-MI results in failure cases (refer to Appx. E), limiting the effectiveness of PPDG-MI. Therefore, we manually select 50 identities

Table 8: Comparison of MI performance with RLB-MI and BREP-MI in the low-resolution setting. The target model M is VGG-16 trained on $\mathcal{D}_{\text{private}}$ = CelebA, GANs are trained on $\mathcal{D}_{\text{public}}$ = CelebA. The symbol ↓ (or ↑) indicates smaller (or larger) values are preferred, and the green numbers represent the attack performance improvement. The running time ratio (Ratio) between prior fine-tuning and MI reflects the relative overhead of fine-tuning.

| Method | Acc@1↑ | Acc@5↑ | KNN Dist↓ | Ratio↓ |
|---|---|---|---|---|
| RLB-MI (black-box) | 38.50 | 65.10 | 1431.93 | / |
| + PPDG-vanilla (ours) | 61.60 (+23.10) | 85.30 (+20.20) | 1337.09 (-94.84) | 0.02 |
| BREP-MI (label-only) | 66.40 | 92.20 | 1171.16 | / |
| + PPDG-vanilla (ours) | 71.10 (+4.70) | 93.10 (+0.90) | 1165.17 (-5.99) | 0.03 |

Table 9: Comparison of MI performance against state-of-the-art defense methods in the low-resolution setting. The target model M is VGG16 trained on $\mathcal{D}_{\text{private}}$ = CelebA, GANs are trained on $\mathcal{D}_{\text{public}}$ = CelebA. **Bold** numbers indicate superior results.

| Method | LOM (GMI) | | KEDMI | | LOM (KEDMI) | |
|---|---|---|---|---|---|---|
| | Acc@1↑ | KNN Dist↓ | Acc@1↑ | KNN Dist↓ | Acc@1↑ | KNN Dist↓ |
| No Def. | 63.19 | 1416.80 | 75.54 | 1297.79 | 84.10 | 1255.15 |
| BiDO-HSIC | 47.71 | 1521.50 | 37.69 | 1547.35 | 68.63 | 1417.85 |
| + PPDG-vanilla | **58.74** | **1455.31** | **44.09** | **1512.79** | **69.68** | **1392.19** |
| NegLS | 25.40 | 1529.62 | 27.15 | 1576.57 | 64.73 | 1320.38 |
| + PPDG-vanilla | **45.44** | **1415.76** | **30.82** | **1532.74** | **68.94** | **1308.09** |

with high-quality samples from 100 identities and present the quantitative results in Tab. 7. Integrating PPDG-vanilla significantly improves MI performance. This improvement can be attributed to the robust selection strategy, which identifies high-quality samples from all pseudo-private data, along with fine-tuning using the original GAN training objectives to enhance density around high-quality pseudo-private samples. These results indicate that even with a different GAN prior (*i.e.,* cGAN), PPDG-MI shows effectiveness with strong compatibility and outperforms SOTA white-box attack.

**Comparison with Black-box MIAs in the low-resolution setting.** In this setting, we use the SOTA RLB-MI [Han et al., 2023] as the baseline for comparison, the results are shown in the upper part of Tab. 8. Han et al. [2023] formulate the latent space search problem as a Markov decision process and solve it using reinforcement learning, which requires tens of thousands of iterations for optimization, leading to inefficiency and a low probability of sampling representative samples. Integrating PPDG-vanilla significantly improves MI performance. This improvement can be attributed to the fine-tuning of the GAN, which effectively enhances the density around the pseudo-private samples, thereby increasing the probability of sampling representative samples. Our experiments demonstrate that PPDG-vanilla boosts the top-1 attack accuracy from $38.50\%$ to $61.60\%$ and the top-5 attack accuracy from $65.10\%$ to $85.30\%$ in the black-box setting, with a substantial reduction in KNN distance from $1431.93$ to $1337.09$. Additionally, the running time ratio indicates a minimal overhead of just $0.02$. These results highlight the effectiveness of our method in improving MI performance in the black-box setting with minimal additional computational cost.

**Comparison with Label-only MIAs in the low-resolution setting.** In this setting, we use the state-of-the-art BREP-MI [Kahla et al., 2022] as the baseline for comparison. The quantitative results are presented in the upper part of Tab. 8. Kahla et al. [2022] introduces a boundary-repelling algorithm to search for representative samples. This algorithm estimates the direction towards the target class's centroid using the predicted labels of the target model over a sphere. Typically, under a radius threshold where the gradient estimator still works reliably, a larger sphere radius indicates a higher-likelihood region around the sphere's centroid. A qualitative illustration of the progression of the reconstructed images towards the actual training images is depicted in Fig. 8. The upper part of Fig. 8 shows the results of BREP-MI, while the lower part shows the results after integrating BREP-MI with PPDG-vanilla. It is evident that with PPDG-vanilla, BREP-MI can find more representative

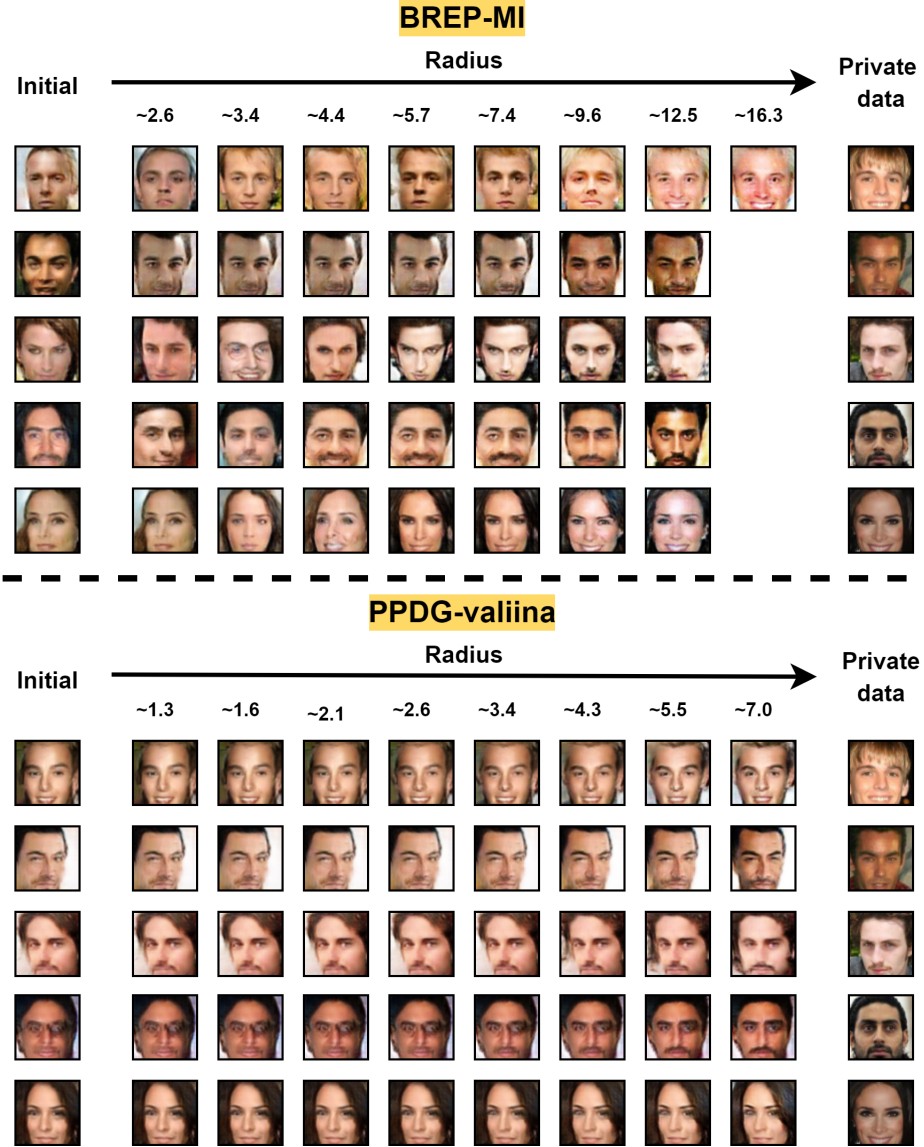

Figure 8: A comparison of the progression of BREP-MI and BREP-MI integrated with PPDG-vanilla from the initial random point to the algorithm's termination, indicating that the latter achieves faster convergence in the search process.

samples at a smaller radius. This demonstrates that our method effectively increases the density of these regions, leading to faster convergence of the search process.

**Attacks against SOTA model inversion defense methods.** In the low-resolution setting, we evaluate SOTA white-box attacks LOM (GMI), KEDMI, and LOM (KEDMI) against SOTA model inversion defense methods, such as BiDO-HSIC [Peng et al., 2022] and NegLS [Struppek et al., 2024]. The results, summarized in Tab. 9 demonstrate that our PPDG-MI consistently outperforms baseline against BiDO-HSIC and NegLS. For instance, when using NegLS as the defense, PPDG-vanilla significantly enhances both the top-1 attack accuracy and the KNN distance metrics. For the LOM (GMI) attack, PPDG-vanilla improves the Acc@1 from 25.40% to 45.44%, and for KEDMI, it increases it from 27.15% to 30.82%. Additionally, PPDG-vanilla shows improvements compared with LOM (KEDMI) by enhancing Acc@1 from 64.73% to 68.94%, and decreasing KNN Distance from 1320.38 to 1308.09. The enhanced MI performance on the target model trained with SOTA defense methods further underscores the effectiveness of PPDG-MI.

Table 10: Ablation study on the number $K$ of high-quality samples selected for fine-tuning. "Time" (seconds per identity) denotes the time required for fine-tuning a single identity .

| Method | K | Acc@1↑ | KNN Dist↓ | Time↓ |
|---|---|---|---|---|
| PPDG-PW | 3 | 80.50 | 0.74 | 149 |
| | 5 | 85.40 | 0.72 | 242 |
| | 7 | 85.65 | 0.72 | 334 |
| | 10 | 86.50 | 0.71 | 470 |
| PPDG-MMD | 5 | 88.30 | 0.70 | 70 |
| | 10 | 89.90 | 0.70 | 119 |
| | 15 | 88.75 | 0.69 | 174 |
| | 20 | 87.60 | 0.69 | 240 |
| | 25 | 86.95 | 0.69 | 298 |
| | 30 | 87.30 | 0.703 | 344 |
| PPDG-CT | 5 | 86.95 | 0.69 | 96 |
| | 10 | 88.15 | 0.69 | 167 |
| | 15 | 87.45 | 0.68 | 238 |
| | 20 | 88.75 | 0.68 | 305 |
| | 25 | 86.60 | 0.68 | 378 |
| | 30 | 86.35 | 0.68 | 451 |

Table 11: Ablation study on fine-tuning different layers of the StyleGAN synthesis network. "Time" (seconds per identity) denotes the time required for fine-tuning a single identity.

| Method | Layers | Acc@1↑ | KNN Dist↓ | Time↓ |
|---|---|---|---|---|
| PPDG-PW | $4^2 - 16^2$ | 67.00 | 0.82 | 237 |
| | $4^2 - 128^2$ | 84.65 | 0.73 | 242 |
| | $4^2 - 1024^2$ | 85.40 | 0.72 | 242 |
| PPDG-MMD | $4^2 - 16^2$ | 73.35 | 0.77 | 111 |
| | $4^2 - 128^2$ | 88.05 | 0.71 | 116 |
| | $4^2 - 1024^2$ | 89.90 | 0.70 | 119 |
| PPDG-CT | $4^2 - 16^2$ | 64.90 | 0.83 | 165 |
| | $4^2 - 128^2$ | 84.20 | 0.71 | 165 |
| | $4^2 - 1024^2$ | 88.15 | 0.69 | 167 |

## D.2 Additional Ablation Study

### D.2.1 MIAs in the High-resolution setting

**Number $K$ of high-quality pseudo-private samples.** The choice of the number $K$ of high-quality pseudo-private samples affects the tuning intensity of the generator G, the informativeness of the empirical local distribution, and the computational cost. In Tab. 10, we illustrate the MI performance for different choices of $K$. As $K$ increases, MI performance initially improves, reaching an optimal point before it starts to decline. This trend can be attributed to two main factors. First, A higher number of samples provides more detailed information about the local data distribution, allowing the generator G to better capture the underlying characteristics of the data. This leads to an initial improvement in MI performance. Second, As $K$ becomes larger, the generator requires more extensive tuning to accommodate the additional samples. This can result in significant changes to the manifold, potentially disrupting the learned data structure and decreasing MI performance. Additionally, larger $K$ values substantially increase computational cost. Thus, considering all these factors, we highlight the $K$ values chosen for various fine-tuning methods in Tab. 10.

**Fine-tune different layers.** Given that the synthesis network in the StyleGAN generator consists of 18 layers—two for each resolution $4^2 - 1024^2$—with earlier layers controlling higher-level features (*e.g.,* general hairstyle, face shape), and later layers controlling more fine-grained features (*e.g.,* finer hairstyle details), we investigate the effect of tuning subsets of these layers. In Tab. 11, we present the results of incrementally adding layers for fine-tuning. Our study reveals that tuning layers with spatial resolutions from $4^2 - 128^2$ achieves comparable results to tuning all layers, *i.e.,* spatial resolutions from $4^2 - 1024^2$. This finding suggests that successful MIAs rely more on inferences about high-level features (*e.g.,* face shape) rather than fine-grained details, aligning with the main goals of MIAs.

### D.2.2 MIAs in the Low-resolution Setting

In this section, we present all ablation studies on MIA in the low-resolution setting to further explore PPDG-MI. The target model, VGG16, and the evaluation model, face.evoLVe, are both trained on CelebA private dataset. GANs are also trained from scratch on CelebA public dataset. Unless otherwise specified, we use KEDMI as the attack method and perform one round of GAN fine-tuning. The size of the pseudo-private dataset is $1,000$, and we fine-tune the GAN for 10 epochs.

Table 12: Ablation study on the number of rounds of fine-tuning.

| Rounds | Acc@1↑ | Acc@5↑ | KNN Dist↓ |
|---|---|---|---|
| Baseline | 58.86 | 85.32 | 1341.36 |
| 1 | 67.55 | 89.24 | 1294.40 |
| 2 | 77.23 | 93.82 | 1237.87 |
| 3 | 80.59 | 94.29 | 1231.55 |
| 4 | 78.07 | 93.32 | 1259.94 |

**Iterative fine-tuning.** Fine-tuning the GAN with generated pseudo-private data increases the probability of sampling data with characteristics closer to the actual private training data. We examine

the impact of iterative fine-tuning on MI performance through experiments with different fine-tuning rounds. We use LOM (GMI) as the attack method and present the results in Tab. 12. The results show that MI performance improves with more fine-tuning rounds, suggesting that pseudo-private data increasingly approximates private training data. However, after the third round, further fine-tuning does not further improve MI performance as observed in earlier rounds. This decline could be due to excessive fine-tuning, which may distort the image manifold and degrade MI performance.

**Impact of the size of the pseudo-private dataset in GAN fine-tuning.** In iterative GAN fine-tuning, the pseudo-private data generated in each round can potentially increase the probability of sampling data with characteristics similar to the real private data in subsequent rounds. This process highlights the importance of determining an appropriate size for the pseudo-private dataset to achieve improved MI performance with an acceptable level of fine-tuning overhead. We investigate how MI performance is affected by varying the size of the pseudo-private dataset, with the results presented in Tab. 13. The results indicate a trend where the MI performance initially improves and

Table 13: Ablation study on the number of pseudo-private data used in GAN fine-tuning, where $|\mathcal{D}^{\mathrm{s}}_{\mathrm{private}}|$ represents the size of the pseudo-private dataset.

| $\|\mathcal{D}^{\mathrm{s}}_{\mathrm{private}}\|$ | Acc@1↑ | Acc@5↑ | KNN Dist↓ |
|---|---|---|---|
| 1000 | 81.99 | 97.36 | 1224.18 |
| 2000 | 84.43 | 98.73 | 1231.73 |
| 3000 | 88.64 | 99.05 | 1222.43 |
| 4000 | 79.65 | 97.57 | 1249.33 |
| 5000 | 78.26 | 98.50 | 1286.53 |

then degrades. This suggests that increasing the size of the pseudo-private dataset up to a certain point (e.g., $3,000$) can enhance MI performance. The observed decline in performance could be attributed to the large amount of pseudo-data adopted, which increases the intensity level of fine-tuning and thus disrupts the image manifold.

**Impact of the number of epochs in GAN fine-tuning.** During iterative GAN fine-tuning, effectively utilizing pseudo-private data while reducing computational overhead is essential. Therefore, the number of tuning epochs is a crucial hyper-parameter. This experiment examines the impact of fine-tuning epochs on MI performance. Tab. 14 shows that MI performance consistently improves as the number of GAN fine-tuning epochs increases, indicating that the GAN effectively learns and utilizes pseudo-private data. However, while MI performance improves, the computational overhead (*i.e.,* fine-tuning time for each identity) increases linearly with the number of epochs.

Table 14: Ablation study on the number of epochs in GAN fine-tuning. "Time" (seconds per identity) denotes the time required for fine-tuning a single identity.

| Epoch | Acc@1↑ | Acc@5↑ | KNN Dist↓ | Time |
|---|---|---|---|---|
| 5 | 75.85 | 95.97 | 1251.84 | 553 |
| 10 | 77.79 | 96.19 | 1245.14 | 1130 |
| 15 | 80.29 | 97.24 | 1243.65 | 1690 |
| 20 | 81.15 | 97.35 | 1235.93 | 2260 |
| 25 | 83.59 | 98.38 | 1220.47 | 2822 |

**Comparison of identity-wise fine-tuning vs. multi-identity fine-tuning.** Compared to fine-tuning the GAN using a single identity, which is specific to one identity and increases overhead, fine-tuning with multiple identities as a whole can significantly reduce computational costs. Thus, we aim to investigate the MI performance of fine-tuning the GAN based on both single-identity and multi-identity approaches, respectively. The re-

Table 15: Ablation study on identity-wise fine-tuning vs. multi-identity fine-tuning.

| Fine-tuning method | Acc@1↑ | Acc@5↑ | KNN Dist↓ |
|---|---|---|---|
| Single-identity | 85.37 | 98.50 | 1207.53 |
| Multi-identity | 81.40 | 94.45 | 1225.42 |

sults, presented in Tab. 15, indicate that the MI performance of multi-identity fine-tuning decreases by $4\%$ compared to single-identity fine-tuning. Despite the reduction in computational overhead, fine-tuning the GAN using multiple identities results in poorer MI performance. This is because fine-tuning with multiple identities induces more intensive changes to the generator.

### D.3 Visualization of Reconstructed Images

In this section, we provide qualitative evidence to demonstrate the effectiveness of our proposed PPDG-MI. Results for high-resolution settings are illustrated in Figs. 9 and 10. Specifically, Fig. 9 presents a visual comparison of reconstructed samples for the first ten identities from three target

models—ResNet-18, DenseNet-121, and ResNeSt-50—trained on the CelebA, using GANs pre-trained on FFHQ. Fig. 10 shows a similar comparison for the same target models trained on the FaceScrub, also using GANs pre-trained on FFHQ. For low-resolution settings, the results are shown in Figs. 11 and 12. Fig. 11 illustrates reconstructed samples for the first ten identities from VGG16 trained on the CelebA private dataset, using GANs trained from scratch on the CelebA public dataset. Fig. 12 depicts reconstructed samples for the first ten identities from VGG16 trained on the CelebA private dataset, using GANs trained from scratch on the FFHQ public dataset.

# E  Discussion

**Scope and Applicability of Model Inversion Attacks.** Model inversion attacks (MIAs) have become a critical area of research in assessing privacy risks, especially for discriminative models that handle sensitive data. While MIAs can be effective in certain data types, tasks, and specific contexts, they also face significant limitations, particularly in cases where the data lacks clear identity markers.

A notable limitation arises in their applicability to biomedical imaging, where MIAs face distinct challenges. For instance, unlike identity-rich data like facial images, chest X-rays generally lack identifiable features that can be easily linked to individuals. This absence of clear personal markers complicates privacy risk evaluation, making it harder to assess the impact of MIAs.

Moreover, even in experimental settings using datasets like ChestXray8 [Wang et al., 2017], where the primary goal is classification tasks such as diagnosing medical conditions, challenges persist. One key concern is whether a reconstructed chest X-ray would represent a generic "average" image or contain identifiable features unique to specific training samples. This issue arises from the inherent complexity of chest X-rays, which often require specialized medical expertise for accurate interpretation.

In summary, MIAs are more effective in domains where data includes explicit identity markers, leading to higher privacy risks as reconstructed images can closely resemble individuals from the training data. However, in fields like biomedical imaging, where data lacks evident identity characteristics, privacy risks are harder to quantify but remain an important area of concern.

**Broader Impacts.** In this paper, we propose a novel model inversion pipeline to enhance the performance of generative model inversion attacks (MIAs), potentially providing new insights and paving the way for future research. From a social perspective, our research on MIAs reveals significant privacy vulnerabilities in machine learning models that, if misused, could compromise sensitive training data. By revealing these risks, we aim to raise awareness and drive the development of robust defense mechanisms and privacy-preserving algorithms that are crucial for enhancing the security of machine learning systems. Overall, although our findings could be misused, the benefits of raising awareness and improving security practices in machine learning systems far outweigh these concerns.

**Failure Case Analysis.** To better assess the feasibility of our proposed method, we closely examined the reconstructed samples. In high-resolution scenarios, we observed that with a single fine-tuning round, PPDG-MI exhibited minimal failures. However, with continued fine-tuning, the visual quality of certain identities significantly deteriorated, indicating substantial alterations in the generator's manifold. Notably, these reconstructed samples still performed well on standard metrics despite poor visual quality, suggesting potential overfitting to these metrics. This finding underscores a limitation in current metrics, such as attack accuracy, which may lack robustness in such cases. Developing more effective and resilient evaluation metrics is an important direction for future research.

In low-resolution scenarios, we observed a decline in visual quality and performance on standard metrics for certain identities after a single round of attack, particularly in the case of LOM (KEDMI) and PLG-MI. We hypothesized that this failure stems from the baseline attack that generates samples with poor visual quality and insufficiently representative features of the target identities. These low-quality samples negatively impact the quality of the samples produced by the fine-tuned generator. With each subsequent fine-tuning round, these negative effects accumulated, progressively degrading the generator's manifold and leading to poorer visualization and MI performance. These findings indicate that the generator's capabilities may be insufficient to produce high-quality samples that positively influence subsequent rounds of GAN fine-tuning, thereby impacting the effectiveness of PPDG-MI. Therefore, utilizing more advanced generators represents a potential direction for better demonstrating the advancements of PPDG-MI.

**Limitations of PPDG-MI.** Conducting model inversion attacks is time-consuming, and the iterative fine-tuning in our proposed PPDG-MI adds further overhead, as shown by the running time ratios between the fine-tuning phase and model inversion phase in our experiments. However, our proposed prior distribution tuning methodology provides a promising solution to mitigate the fundamental distribution discrepancies between private and prior distributions. We hope our work will inspire future research to develop more efficient and effective tuning methods.

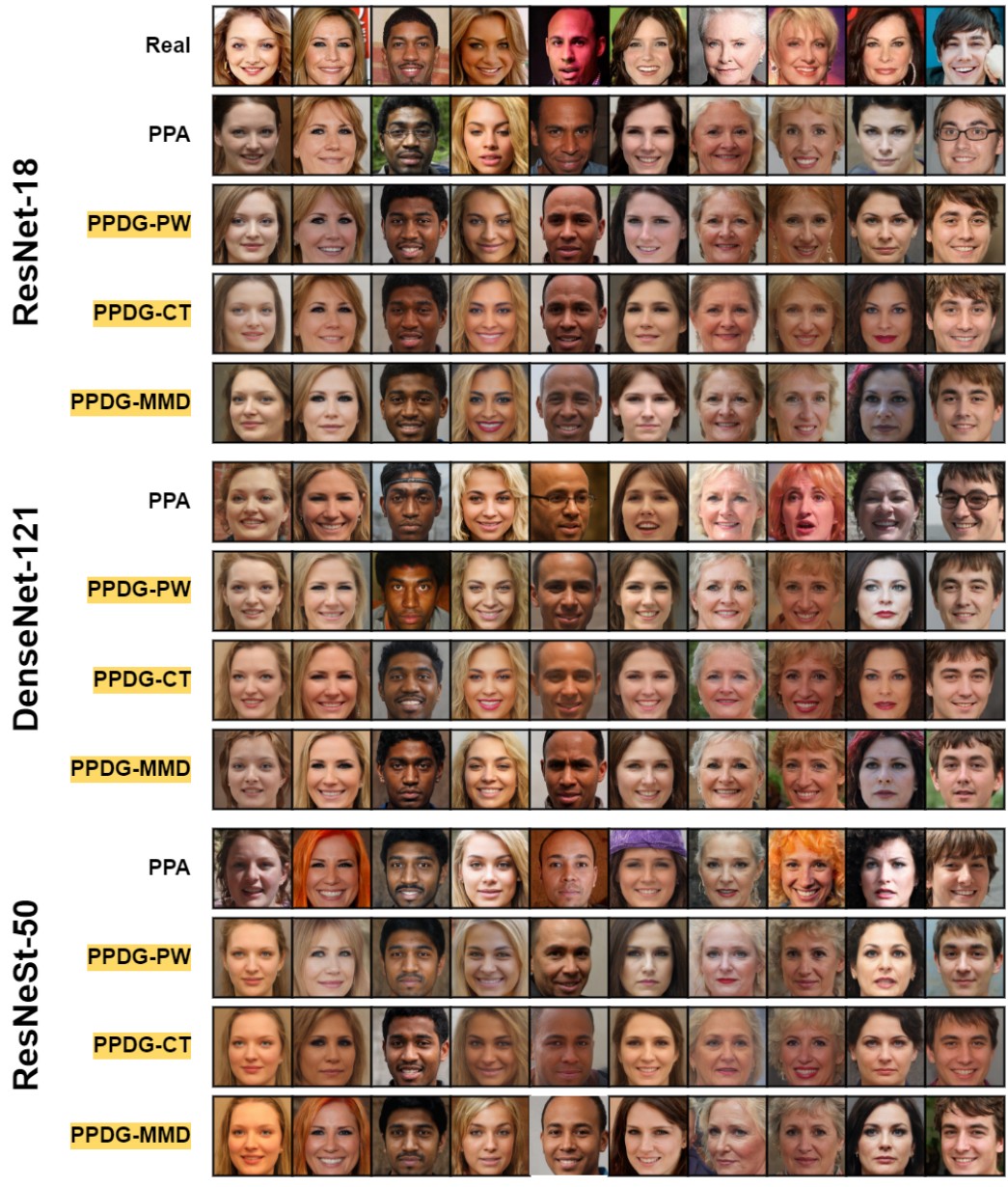

Figure 9: Visual comparison in high-resolution settings. We illustrate reconstructed samples for the first ten identities in $\mathcal{D}_{\text{private}}$ = CelebA using GANs pre-trained on $\mathcal{D}_{\text{public}}$ = FFHQ.

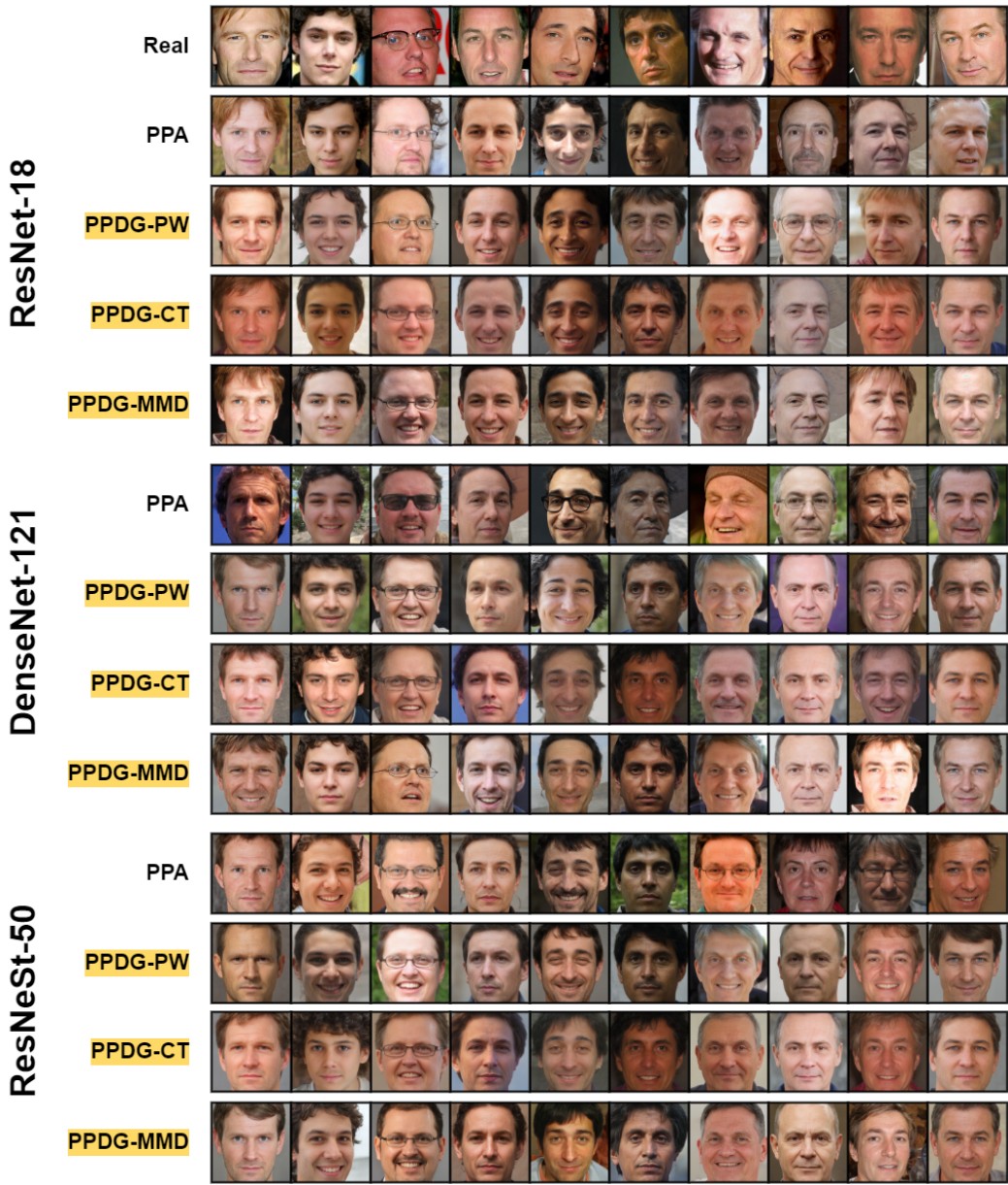

Figure 10: Visual comparison in high-resolution settings. We illustrate reconstructed samples for the first ten identities in $\mathcal{D}_{\text{private}}$ = FaceScrub using GANs pre-trained on $\mathcal{D}_{\text{public}}$ = FFHQ.

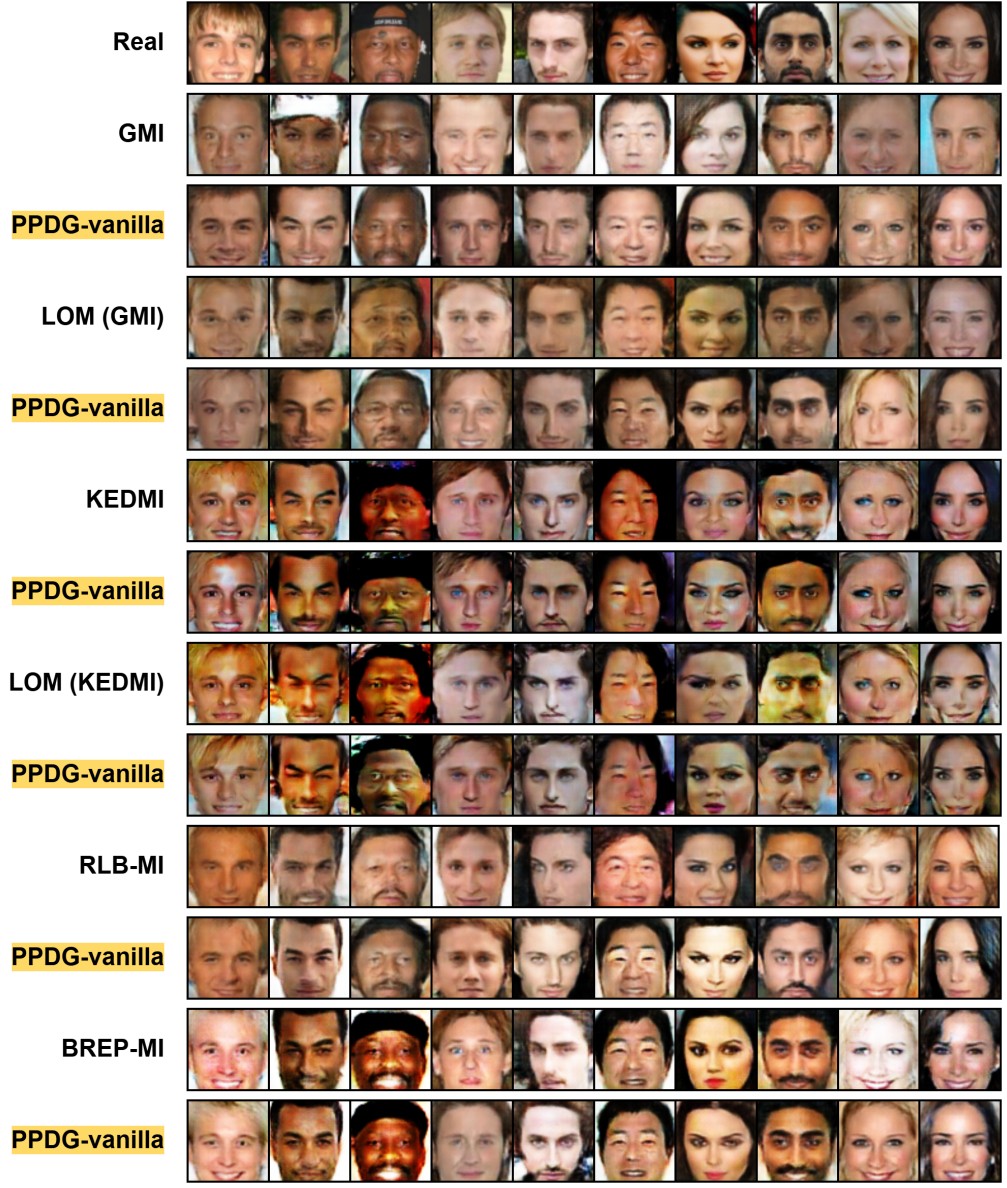

Figure 11: Visual comparison in low-resolutions settings. We illustrate reconstructed samples for the first ten identities in $\mathcal{D}_{\text{private}}$ = CelebA using GANs trained from scratch on $\mathcal{D}_{\text{public}}$ = CelebA.

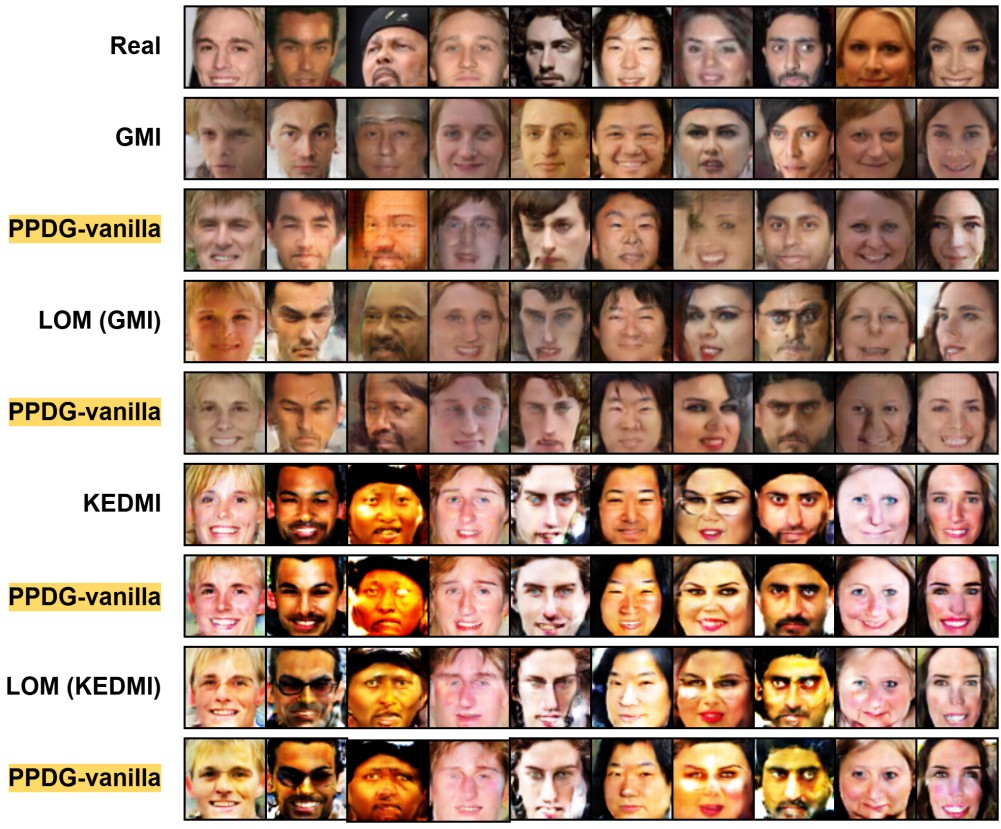

Figure 12: Visual comparison in low-resolutions settings. We illustrate reconstructed samples for the first ten identities in $\mathcal{D}_{\text{private}}$ = CelebA using GANs trained from scratch on $\mathcal{D}_{\text{public}}$ = FFHQ.

