# OpenReview forum: "Pseudo-Private Data Guided Model Inversion Attacks"
_NeurIPS.cc/2024/Conference — NeurIPS 2024 poster_

### Official Review · Reviewer_eDgc · 2024-06-14

**Soundness:** 3
**Presentation:** 3
**Contribution:** 3
**Rating:** 7
**Confidence:** 5

**Summary:**

The paper introduces a novel method to enhancing model inversion attacks (MIAs), which aim to reconstruct class characteristics from a trained classifier. Typically, MIAs rely on training image priors, such as GANs, on public data that differ in distribution from the target model's training data. This distributional discrepancy reduces the effectiveness of MIAs in accurately reconstructing features of the target class. To address this issue, the paper presents pseudo-private data-guided MIAs. The proposed method initially conducts existing MIAs to gather a set of attack samples that reveal features of the target classes. A subset of high-quality samples is selected based on robust prediction scores under the target model. The GAN's generator is then fine-tuned on this subset to increase the sampling density around the pseudo-private data. Subsequent attacks using the updated generator show a significant improvement in attack results across various attacks and settings.

**Strengths:**

- Existing MIAs either focused on the initial training of GANs or the attack's optimization method. This paper takes another direction and adds the idea of fine-tuning the GAN's generator on attack results from a previous run. This method adds a novel and exciting dynamic dimension to the literature on model inversion attacks.
- The approach is well-motivated and supports its claims empirically with toy examples and results on standard MIA benchmarks. This fact makes the proposed improvements convincing and reasonable. Particularly Sec. 3.3 adds valuable insights to the methodology.
- The evaluation investigates a broad range of model architectures, datasets, and types of MIAs. The proposed method shows improved results across all settings, supporting the paper's claims and the method's efficiency.

Small remark:
- The appendix is well-formatted and clearly arranged. It provides valuable additional insights.

**Weaknesses:**

- The evaluation is missing some aspects. For the investigated attacks, it would be interesting to compare the results to PLG-MI [1], which takes a similar direction (training a conditional GAN on pseudo-labeled data to decouple the latent space for different target classes) and shows strong attack results.
- Also, the evaluation includes no investigation of existing defense methods like MID [2], BiDO [3], and negative label smoothing [4]. For MID and BiDO, existing attacks have already (partly) broken the defenses; see [4,5]. However, the authors [4] argue that negative label smoothing limits the information provided by a model's prediction scores, which, in turn, might affect the subset selection process used by the pseudo-private data-guided MIAs proposed method. It would be interesting to see if the proposed attack improvement also breaks this type of defense.
- The evaluation focuses on metrics computed on an evaluation model trained on the same data as the target model has been. While this is a common approach in MIA literature, it might provide some misleading results. For example, given that the attacks might reconstruct adversarial features instead of actual robust and characteristic class features, the fine-tuned GAN might also generate images containing adversarial features. Since we know from the literature that adversarial features can be transferable, the attack results could similarly fool the evaluation model and, therefore, overestimate an attack's success. Therefore, it might be reasonable to include other metrics, e.g., the FaceNet distance used in the PPA paper. Another option to measure the attack's success could be the information extraction score introduced in [4]. Using the FaceNet distance in addition to the Attack Accuracy in Fig. 5 would further support the paper's claims.
- Giving more intuition on why MMD and CT are used instead of, e.g., a KL Divergence would improve the paper. Similarly, why use LPIPS as a similarity metric instead of another type of metric, e.g., FaceNet distance?

Minor remarks:
- The term "MMD" used in the caption of Fig. 1 should be introduced before referring to this figure. Also, "MI" (L39) is not formally introduced. It probably stands for "model inversion". However only "MIA" is introduced, which might lead to confusion of the reader.
- Section 2.2: There should be some motivation for why MMD and CT are introduced here. It makes sense later in the paper, but these concepts appear a bit surprising at the point of the background section. 1-2 introduction sentences would support the motivation here. Also, intuition about what MMD and CT are measuring and how they differ supports understanding these concepts.
- L132: From my understanding, the "linear interpolation" between both distributions means that the sample share $\alpha$ is sampled from X_prior and $(1-\alpha)$ is sampled from X_private. This should be clarified in the writing.
- Fig. 3: The font size in the legend is too small and should be increased. Also, the size of the data points should be increased, particularly the red ones in the final image. Currently, those data points are hard to see in (b).
- To make the approach more clear, think about adding "Step-4: Repeating the attack with the updated generator". Currently, the approach only describes the steps of fine-tuning the generator, and it might not be clear to all readers that the attacks will be repeated after this step.
- L296: The figure reference is broken, it currently states "Fig. 4.3" instead of "Fig. 5".

Overall, I liked the paper and the mentioned weaknesses can probably be addressed during the rebuttal. Therefore, I encourage the authors to participate in the discussion and will increase my score if the mentioned weaknesses are sufficiently addressed.

[1] Yuan et al. "Pseudo Label-Guided Model Inversion Attack via Conditional Generative Adversarial Network". AAAI 2023
[2] Wang et al. "Improving Robustness to Model Inversion Attacks via Mutual Information Regularization". AAAI 2021
[3] Peng et al. "Bilateral Dependency Optimization: Defending Against Model-inversion Attacks". KDD 2022
[4] Struppek et al. "Be Careful What You Smooth For: Label Smoothing Can Be a Privacy Shield but Also a Catalyst for Model Inversion Attacks. ICLR 2024
[5] Nguyen et al. "Re-thinking Model Inversion Attacks Against Deep Neural Networks". CVPR 2023

**Questions:**

- There is the risk of a mode collapse in fine-tuning the generator. Does the approach require regularization to avoid this failure case, or is the method already robust enough to avoid it?
- Regarding the Ratio metric: From my intuition, the ratio should always be > 2, because the attacks take at least twice the time compared to the baseline (1x runnning the baseline + 1x repeating the attack with the adjusted generator). A more detailed description of this metric would improve the understanding.

**Limitations:**

Some limitations are discussed in Appx. E. However, I think the limitation section could include some failure cases and additional limitations of the method. In which settings did it fail? Are there any additional requirements for the method to work?

While the paper proposes a novel type of attack, I do not think there will be negative societal impact, given that there already exists a long list of publicly available MIA literature and implementations.

---

> ### Author Rebuttal · Authors · 2024-08-07
>
> Sincerely thank you for your constructive comments and generous supports! Please see our detailed responses to your comments and suggestions below.
>
> > W1, W2: Missing evaluations on PLG-MI and state-of-the-art model inversion defenses.
>
> Regarding these additional evaluations, please refer to the general response. These results will be included in the main results section (i.e., Section 4.2) of the final version of our paper.
>
> > W3. The evaluation focuses on metrics computed on an evaluation model trained on the same data as the target model ... further support the paper's claims.
>
> We appreciate the reviewer's insightful comments regarding the potential limitations of using an evaluation model trained on the same data as the target model. **We have indeed considered this concern in the paper**. For the PPA-based experiments, to mitigate the risk of misleading results, particularly with regard to adversarial features, we calculated the KNN Dist metric using the penultimate layer of a pre-trained FaceNet model, as you suggested. Details of this setup are provided in Appendix C.5 of the paper.
>
> Additionally, in response to concerns about Fig. 5, we have computed the FaceNet distance as an additional evaluation metric, with results as follows:
>
> | Method           | Round                 | Acc@1$\uparrow$ | KNN Dist $\downarrow$   |
> |------------------|-----------------------|-----------------|------------|
> | PPDG-PW          | 0                     | 59.10           | 0.8559     |
> |                  | 1                     | 83.15           | 0.7082     |
> |                  | **2**                 | **89.42**       | **0.6824** |
> | PPDG-MMD         | 0                     | 59.10           | 0.8559     |
> |                  | 1                     | 88.53           | 0.6795     |
> |                  | **2**                 | **95.12**       |**0.6313**  |
> | PPDG-CT          | 0                     | 59.10           | 0.8559     |
> |                  | 1                     | 87.32           | 0.6754     |
> |                  | **2**                 | **93.03**       | **0.6397** |
>
> | Method           | Data Selection | Acc@1$\uparrow$ | KNN Dist $\downarrow$   |
> |------------------|----------------------------------------|-----------------|-----------|
> | PPDG-PW          | Random samples                         | 77.45           | 0.7566    |
> |                  | ****High-quality samples****           | **85.40**       | **0.7233**|
> | PPDG-MMD         | Random samples                         | 85.10           | 0.7200    |
> |                  | ****High-quality samples****           | **89.90**       | **0.6981**|
> | PPDG-CT          | Random samples                         | 84.65           | 0.7053    |
> |                  | ****High-quality samples****           | **88.15**       | **0.6866**|
>
> | Method           | Discriminator         | Acc@1$\uparrow$ | KNN Dist $\downarrow$   |
> |------------------|-----------------------|-----------------|------------|
> | PPDG-PW          | w/o | 63.70           | 0.8391     |
> |                  | **w/**  | **85.40**        | **0.7233** |
> | PPDG-MMD         | w/o | 76.40           | 0.7576     |
> |                  | **w/**  | **89.90**        | **0.6981** |
> | PPDG-CT          | w/o | 73.20           | 0.7860     |
> |                  | **w/**  | **88.15**       | **0.6866** |
>
> These results are consistent with the Attack Accuracy metric, further supporting the effectiveness of our method.
>
> > W4. Giving more intuition on why MMD and CT are used ... FaceNet distance?
>
> - We chose MMD and CT over KL divergence primarily due to the inherent limitations of KL divergence, which requires the two probability distributions to have the same support and is often inapplicable when one or both distributions are implicit with unknown probability density functions (PDFs) [r1, r2]. In contrast, MMD and CT do not have these limitations. They are also amenable to mini-batch based optimization and are straightforward to implement in practice.
> - Regarding the use of LPIPS as a similarity metric, we initially experimented with FaceNet distance, using the penultimate layer features of a pre-trained FaceNet model to measure similarity. However, we found that the results were not satisfactory, possibly because using features from a single layer did not provide sufficient discriminative semantic information. Therefore, we adopted the approach outlined in the original StyleGAN2 paper [r3], using LPIPS, which measures similarity based on a concatenation of multiple hidden layer representations from a VGG feature extractor. This method captures more comprehensive semantic information, making LPIPS more discriminative and, consequently, more effective than FaceNet distance for our purposes.
>
> W5. (Minor remarks):
>
> > W5.1. The term "MMD" used in the caption of Fig. 1 should be introduced ... lead to confusion of the reader.
>
> Thank you for your helpful suggestions. We have updated the caption of Fig. 1 to include the full term for the abbreviation "MMD," which stands for maximum mean discrepancy. Additionally, we have clarified the term "MI" on line 39 by introducing its full name, "model inversion (MI)," to avoid any confusion for the readers.

---

> ### Author Response · Authors · 2024-08-07
> **Remaining Responses to Reviewer eDgc (1/2)**
>
> > W5.2. Section 2.2: There should be some motivation for why MMD and CT are introduced here ... understanding these concepts.
>
> Thank you for your valuable suggestions. We have revised Section 2.2 to include additional context explaining the motivation for introducing MMD and CT at this point in the paper. Specifically, we added 1-2 introductory sentences to clarify the relevance of these measures to our method. Additionally, we included a brief explanation of what MMD and CT measure and how they differ, providing readers with a clearer understanding of these concepts. The specific content added is as follows:
>
> "To effectively align distributions in our subsequent methods, it is essential to introduce metrics that can accurately quantify the differences between them. Two commonly used measures for this purpose are maximum mean discrepancy (MMD) and conditional transport (CT). MMD focuses on mean differences using kernel methods, while CT incorporates cost-based transport distances, offering complementary perspectives on distributional discrepancies."
>
> > W5.3. L132: From my understanding, the "linear interpolation" between both distributions means that the sample share $\alpha$ is sampled from X_prior and $(1-\alpha)$ is sampled from X_private. This should be clarified in the writing.
>
> Thank you for your feedback. We have made the necessary clarifications in the manuscript. The specific content added is as follows:
>
> "To evaluate the impact of this distribution discrepancy on MI performance, we create a series of proxy prior distributions through linear interpolation, where a mixing coefficient $\alpha\in [0,1]$ determines the proportion of samples drawn from each distribution. Specifically, a fraction $\alpha$ of samples is drawn from $\mathrm{P}(\mathcal{X}{\text{prior}})$, and the remaining $(1-\alpha)$ is drawn from $\mathrm{P}(\mathcal{X}{\text{private}})$."
>
>
> > W5.4. Fig. 3: The font size in the legend ... Currently, those data points are hard to see in (b).
> > W5.6. L296: The figure reference is broken, it currently states "Fig. 4.3" instead of "Fig. 5".
>
> Thank you for your detailed observations. We have made the necessary adjustments based on your suggestions. Specifically, we increased the font size in the legend and enlarged the data points in Fig. 3 to enhance their visibility and make them easier to interpret. Additionally, we corrected the figure reference on line 296, updating it from "Fig. 4.3" to "Fig. 5" as appropriate.
>
> > W5.5. To make the approach more clear, think about adding "Step-4: Repeating the attack with the updated generator" ... be repeated after this step.
>
> Thank you for your suggestion. As clarified in Section 3.2, PPDG-MI consists of three **iterative** steps, so adding a separate "Step-4: Repeating the attack with the updated generator" may be redundant. However, to enhance clarity, we will add a note at the end of Step-3 indicating "return to Step-1 and repeat the attack with the updated generator." This addition should make the iterative nature of the process more explicit to all readers.
>
> > Q1. There is the risk of a mode collapse in fine-tuning the generator. Does the approach require regularization to avoid this failure case, or is the method already robust enough to avoid it?
>
> - For MIAs focusing on low-resolution tasks, we adopt a principled tuning strategy, fine-tuning $\mathrm{G}$ and $\mathrm{D}$ using the original GAN training objective on $\mathcal{D}\_{\text{public}} \cup \mathcal{D}\_{\text{private}}^{\text{s}}$. This approach mitigates the risk of mode collapse.
> - However, for MIAs targeting high-resolution tasks, such as PPA [r4], we are unable to apply a principled tuning strategy due to the lack of access to the GAN training specifics. Instead, we employ an empirical strategy to fine-tune $\mathrm{G}$. While this approach may affect the quality of the generated images, it does not lead to mode collapse, as we only make slight alterations to the generator $\mathrm{G}$. Currently, we do not employ regularization to avoid this failure case but instead manage it by controlling the fine-tuning strength (i.e., through hyperparameter adjustment). We believe that incorporating regularization could further enhance the robustness of this process.

---

> ### Author Response · Authors · 2024-08-07
> **Remaining Responses to Reviewer eDgc (2/2)**
>
> > Q2. Regarding the Ratio metric.
>
> Thank you for your insightful question. To ensure a fair comparison, we maintained **an equal number of queries** to the target model $\mathbf{M}$ during the inversion process for both the baseline PPA and PPDG-MI. For example, in the experiment where $\mathcal{D}_{\text{public}}$ = FFHQ and $\mathrm{M}$ = ResNet-18, the baseline attack's optimization iterations were set to 70, while PPDG-MI was configured with 35 optimization steps per round. Due to space constraints, we have detailed these attack parameters in Appendix C.4.
>
> Additionally, after considering your perspective, we agree that maintaining the same number of optimization iterations per round as the baseline is a more reasonable approach to validate the effectiveness of PPDG-MI. This setup would only further enhance our experimental results. We appreciate your feedback and will consider this for futher experiments.
>
> > Limitations: The limitation section could include some failure cases and additional limitations of the method.
>
> We appreciate the reviewer's suggestion. We will further examine the experimental results and include a discussion of failure cases and additional limitations of the method in the final version of our paper.
>
> ---
> **References**:
>
> [r1] Tran et al. "Hierarchical Implicit Models and Likelihood-Free Variational Inference." In NeurIPS, 2017.
>
> [r2] Yin et al. "Semi-Implicit Variational Inference." In ICML, 2018.
>
> [r3] Karras et al. "Analyzing and Improving the Image Quality of StyleGAN." In CVPR, 2020.
>
> [r4] Struppek et al. "Plug & Play Attacks: Towards Robust and Flexible Model Inversion Attacks." In ICML, 2022.

---

> > ### Comment · Reviewer_eDgc · 2024-08-09
> >
> > Dear authors,
> >
> > after reading all reviews and responses, I decided to slightly increase my score since my main weaknesses (missing evaluations, baselines, defenses, clarifications) have been addressed. I think the authors did a good job at providing additional information requested by the reviewers. However, I think the initial submission is missing too many details and experiments to justify a higher score.

---

> > > ### Author Response · Authors · 2024-08-09
> > > **Thank you for your constructive comments**
> > >
> > > We would like to thank you again for your time and efforts in reviewing our paper, as well as for your generous support of our work. Your insightful comments have greatly improved the quality of the paper.
> > >
> > > Sincerely,
> > >
> > > The Anonymous Authors

---

### Official Review · Reviewer_qx3K · 2024-06-15

**Soundness:** 3
**Presentation:** 3
**Contribution:** 1
**Rating:** 4
**Confidence:** 4

**Summary:**

This work introduces a novel application of a generative model inversion attack utilising dynamic (pseudo-private) priors, improving the existing results of MI.

**Strengths:**

The work is very clear, the idea itself makes sense and the results are well-presented. Authors explicitly target a specific problem and manage to outperform the existing works in the MI field. The method itself is well-motivated and evaluated in a variety of settings.

**Weaknesses:**

So while the idea is straightforward and makes a lot of sense in the chosen setting, the reliance on priors in MI is a) hardly novel [1,2] and b) comes with caveats not covered in the paper. I appreciate that there are various formulations of MI in literature and the one used in this work is generative MI specific, but the main principle of relying on a dynamic prior has previously been covered in the context of MIs.

With respect to point b): if the generative model is conditioned on the pseudo-generations, I suspect this can lead to the issues of bias and/or mode collapse. What I mean here is that there is no guarantee that the reconstructions obtained as part of the pseudo-generation process would resemble the 'actual' training data. This is problem number one, but lets assume the images generated in this step are valid samples from the training dataset. The fact that you are able to generate them means (similarly to the results of the Secret revealer discussed in this work) that they were 'easier' samples with respect to MI vulnerability. And currently, there is no reason for me to believe that this would help you reconstruct the more 'difficult' or informative samples (often on the tails of the distribution [3]). So to simplify: I am not convinced that by leveraging the pseudo-private samples you would be able to improve the attack results meaningfully, as now you would condition your reconstructions to those that are of high similarity to the pseudo-private ones (and I would argue, making it close to impossible to now reconstruct the samples that do not fall under this category, which you could have done with a frozen decoder which would not discriminate between these).

One step which is mentioned as 'optional' is selection of samples that are meaningful. What does this imply? To me this is a very important (and an incredibly challenging) task, which was not given enough discussion in the manuscript, given that you are conditioning your further inversions based on the quality of the pseudo-generated ones. How do you effectively measure it to avoid collapsing into the region of pseudo-reconstructions alone (i.e. similarly to my comments above, how do you measure and use this quantity to encourage MI diversity)?

While the focus of the work is clearly computer vision, it does limit the impact of the findings to these specific settings, making this more of an incremental improvement than a novel paradigm of MI, which the paper positions itself to be.

Minor: the convention is typically MI for model inversion, rather than MIA (which is often used for membership inference), making it a bit more difficult to follow.

[1] - Hatamizadeh, Ali, et al. "Do gradient inversion attacks make federated learning unsafe?." IEEE Transactions on Medical Imaging (2023).
[2] - Usynin, Dmitrii, Daniel Rueckert, and Georgios Kaissis. "Beyond gradients: Exploiting adversarial priors in model inversion attacks." ACM Transactions on Privacy and Security 26.3 (2023): 1-30.
[3] - Feldman, Vitaly. "Does learning require memorization? a short tale about a long tail." Proceedings of the 52nd Annual ACM SIGACT Symposium on Theory of Computing. 2020.

**Questions:**

Could you explain why are you making a connection to adversarial samples in the background? Sure they possess the features you describe, but how is this relevant for MI discussion?

The difference between the low/high dimensional reconstructions sounds rather artificial (it seems this is just pre-trained vs FT GAN), is it really necessary to separate these? Are there are other differences making this separation clearer?

**Limitations:**

As per weaknesses above: to me this seems like an incremental improvement in a relatively niche area, which is more suited to a conference specializing in attacks on ML or PPML, as these results are constrained to specific ML settings.

---

> ### Author Rebuttal · Authors · 2024-08-07
>
> Thank you for your time in reviewing our work and for your constructive comments. Please see our detailed responses to your comments and suggestions below.
>
> > W1. So while the idea is straightforward and makes a lot of sense ... in the context of MIs.
>
> First, we would like to clarify that the problem investigated in our paper focuses on model inversion attacks, where the adversary seeks to recover private training data by exploiting access to **only a well-trained target model $\mathrm{M}$**. In this context, the adversary is limited to querying $\mathrm{M}$ and possesses knowledge of the target data domain but lacks specific details about $\mathcal{D}_{\text{private}}$. This is fundamentally different from the two papers you cited, which investigate gradient inversion attacks. In gradient inversion attacks, the adversary has access to data gradients; however, this information is not available in our case.
>
> Regarding the novelty of our work, we do not claim that reliance on a generative prior is our primary contribution. Rather, this approach was first proposed by GMI [r1]. However, **our work is the first to introduce the use of a dynamic prior**, whereas previous studies [r1, r2, r3] in generative MIAs have relied on a fixed generative prior.
>
> > W2.1. With respect to point b): if the generative model is conditioned on the pseudo-generations ... 'actual' training data.
>
> We would like to emphasize that **generative model inversion attacks have demonstrated the ability to recover samples that closely resemble actual training data [r2, r3]**, by leveraging advanced model inversion optimization techniques and strong image priors.
>
> The ability to recover such samples stems from the fact that the well-trained target model has learned discriminative features specific to the class. The effectiveness of these attacks is a key reason why this area has garnered increasing attention, as it underscores the privacy leakage risks inherent in machine learning models.
>
> > W2.2. ...but lets assume the images generated in this step are valid samples ... high similarity to the pseudo-private ones.
>
> We would like to clarify a few points regarding your concerns. First, the distinction between "easier" and "more difficult" or informative samples is not as clear-cut in the context of model inversion attacks, especially when using balanced datasets, which are commonly employed in this domain. For instance, private training datasets like FaceScrub consist of 106,863 face images from 530 celebrities, with about 200 images per person, and the CelebA subset contains 30,000 face images of 1,000 identities, with about 30 images per person.
>
> In generative model inversion attacks, **the underlying assumption is that the model has already learned discriminative features between classes**; otherwise, the task would be impossible to accomplish. As long as the well-trained target model captures these discriminative features and the discrepancy between the prior distribution and the private data distribution is small, representative images reflecting these features can be recovered with generative model inversion techniques. These representative images are sufficient to reveal privacy-sensitive information related to the training data.
>
> Under this assumption, the primary challenge is to minimize the discrepancy between the prior distribution and the private data distribution, thereby increasing the likelihood of sampling actual training data (cf. right panel of Fig. 2). The dynamic prior method we propose specifically aims to reduce this discrepancy. Furthermore, both qualitative and quantitative experimental results have demonstrated the effectiveness of our method, i.e., it can recover samples that more closely resemble actual training data.
>
>
> > W3. One step which is mentioned as 'optional' is selection of samples that are meaningful. What does this imply? ...
>
> We appreciate the reviewer's insightful question. Step-2 is labeled as "optional" to reflect the differences between low-resolution MIAs and high-resolution MIAs at this stage.
>
> Specifically, for low-resolution MIAs, we adopt a principled tuning strategy. In this case, we fine-tune $\mathrm{G}$ and $\mathrm{D}$ using the original GAN training objective on $\mathcal{D}\_{\text{public}}$ (e.g., 30,000 samples) and $\mathcal{D}\_{\text{private}}^{\text{s}}$ (e.g., 1,000 samples per identity). We utilize **all** pseudo-private data because some MIA algorithms involve highly time-consuming optimization processes (e.g., black-box MIAs), and effective density enhancement requires a sufficient amount of pseudo-private data. Additionally, by incorporating $\mathcal{D}\_{\text{public}}$ during fine-tuning, we mitigate the risk of mode collapse.
>
> For high-resolution MIAs, we propose a tuning strategy that leverages only the high-quality pseudo-private dataset $\mathcal{D}\_{\text{private}}^{\text{s}'}$, due to the high memory consumption during optimization, which necessitates selecting a subset of high-quality samples (e.g., 10 out of 100 samples). While this approach may affect the quality of the generated images, it does not lead to mode collapse. We mitigate this risk by carefully adjusting the hyperparameters, resulting in only slight alterations to the generator $\mathrm{G}$.
>
> To address any confusion, we will remove the "optional" label and clarify the differences in pseudo-private data selection during Step-2 in the attack parameters section (Appendix C.4).

---

> ### Author Response · Authors · 2024-08-07
> **Remaining Responses to Reviewer qx3K**
>
> > W4. While the focus of the work is clearly computer vision ... which the paper positions itself to be.
>
> Model inversion attacks have garnered increasing attention in the trustworthy machine learning area due to their potential to reveal privacy risks in machine learning models. While current generative MIAs primarily focus on either the initial training process of GANs or the optimization techniques used in the attacks, our paper takes a different direction. We introduce a novel method by fine-tuning the GAN's generator based on the attack results from previous runs.
> This approach introduces a dynamic and iterative dimension to model inversion attacks, **expanding the current understanding and application of generative MIAs**. Although our work is primarily focused on computer vision, **the underlying principles and methodologies could potentially be adapted to other domains, making this more than just an incremental improvement**.
>
> > W5. Minor: the convention is typically MI for model inversion, rather than MIA (which is often used for membership inference), making it a bit more difficult to follow.
>
> Indeed, both "MI attacks" and "MIAs" have been used to refer to model inversion attacks in the literature. For example, "MI attacks" is used in works such as GMI [r1] and LOMMA [r3], while "MIAs" is the terminology adopted in PPA [r2] and LS [r4].
>
> > Q1. Reason for making a connection to adversarial samples in the background.
>
> Thank you for highlighting this point. Traditional MIAs [r5] on DNNs trained with image data adopt direct optimization in the input space, which can lead to the generation of adversarial samples. This limitation led Zhang et al. [r1] to propose generative MIAs as a more effective alternative. We have outlined this progression in the introduction section, specifically in lines 26-41. To eliminate any confusion, we will make the necessary clarifications in the problem setup section.
>
> > Q2. The difference between the low/high dimensional reconstructions sounds rather artificial (it seems this is just pre-trained vs FT GAN), is it really necessary to separate these? Are there are other differences making this separation clearer?
>
> We appreciate the reviewer's constructive comments. The distinction between low-resolution and high-resolution settings is based on the typical approach in each scenario: the low-resolution setting usually involves training GANs from scratch using low-resolution public auxiliary data, while the high-resolution setting involves using pre-trained StyleGAN models trained on high-resolution data. This separation was originally introduced by the state-of-the-art model inversion defense LS [r4], and we followed this established setup. We acknowledge your point and will consider your suggestion to categorize the approaches based on whether the GAN is trained from scratch or a pre-trained StyleGAN is used.
>
> ---
> **References**:
>
> [r1] Zhang et al.  "The Secret Revealer: Generative Model-inversion Attacks Against Deep Neural Networks." In CVPR, 2020.
>
> [r2] Struppek et al. "Plug & Play Attacks: Towards Robust and Flexible Model Inversion Attacks." In ICML, 2022.
>
> [r3] Nguyen et al. "Re-thinking Model Inversion Attacks Against Deep Neural Networks
> ." In CVPR, 2023.
>
> [r4] Struppek et al. "Be Careful What You Smooth For: Label Smoothing Can Be a Privacy Shield but Also a Catalyst for Model Inversion Attacks." In ICLR, 2024.
>
> [r5] Fredrikson et al. "Model Inversion Attacks that Exploit Confidence Information and Basic Countermeasures." In CCS, 2015.

---

> > ### Comment · Reviewer_qx3K · 2024-08-10
> > **Response to the rebuttal**
> >
> > I would like to thank the authors for their comprehensive response. While some issues were clarified, I am still not convinced about a number of points.
> >
> > > our work is the first to introduce the use of a dynamic prior, whereas previous studies [r1, r2, r3] in generative MIAs have relied on a fixed generative prior.
> >
> > I understand the message the authors are trying to convey, but I would argue that both the r1 and the r2 pick the prior dynamically (i.e. there is a certain heuristic used to select a prior to optimise for, so its not 'static' in the sense the authors seem to suggest). But this is more of a discussion on the setting and the context, not this work itself.
> >
> > > generative model inversion attacks have demonstrated the ability to recover samples that closely resemble actual training data
> >
> > This specific point I do not disagree with per se, but the issue here is that determining the privacy violation of data which 'closely resembles' the training data is not straightforward. Were you to run this attack in, lets say, a biomedical domain - you may be able to reconstruct a chest X-ray scan learnt by the model. But you are likely unable to determine a) if there is a specific patient this scan corresponds to, b) if there are any features which actually belong to the training record that you attempted to reconstruct and are not just an 'average looking chest X-ray'. The same logic, in my view, applies to the setting discussed in this work: while the idea of using priors (and particularly dynamic priors) does make sense in some settings, where potential over-conditioning is not a major issue (e.g. one may argue that many facial features may be similar to one another), this is not the case universally. And while on its own this does not in any way diminish the contributions of this work, I do not believe that this is something that can be easily applied in other domains.
> >
> > The aforementioned issue could have been eliminated should you demonstrate superior performance of your method in a distinctly different setting, for instance (even within the imaging modality, but ideally beyond that). Therefore I am inclined to keep my score unchanged: I am still not convinced by the scope of the novel contribution proposed in this work.

---

> > > ### Author Response · Authors · 2024-08-11
> > > **Response to Reviewer qx3K (2/2)**
> > >
> > > > Re: The issue here is that determining the privacy violation of data which 'closely resembles' the training data is not straightforward.
> > >
> > > Regarding the applicability of our methods to 'other domains,' our intention was not to imply that our approach is directly transferable across all data domains. Instead, we aimed to suggest that the use of dynamic priors could be advantageous in other tasks where a frozen generator are utilized, such as in image editing [r10]. We hope this clarification addresses the concerns raised.
> > >
> > > We appreciate the reviewer’s insightful comments regarding the application of generative model inversion attacks, particularly in biomedical imaging domains such as chest X-rays. We agree that privacy concerns in these domains are more challenging to assess due to the less distinctive nature of chest X-rays compared to identity-bearing data like facial images.
> > >
> > > We emphasize that **while model inversion attacks are not universally applicable for evaluating all types of privacy leakage, they remain a crucial direction for understanding and measuring privacy risks**, especially given that model inversion attacks can potentially reconstruct training data with just access to a well-trained classifier. This has been extensively studied in the literature [r1-r3, r6-r9] and is recognized as a trending area of research due to its significant implications, particularly in applications like facial recognition, where the attacks can achieve highly accurate reconstructions.
> > >
> > > To further clarify the specific case you mentioned:
> > >
> > > Regarding the first point—whether a reconstructed sample corresponds to a specific patient—it is indeed true that chest X-rays typically lack strong identity markers that can be directly linked to an individual. In the context of model inversion attacks, it is uncommon to associate a specific chest X-ray with a particular patient, as chest X-rays do not possess distinctive identity characteristics like facial images. Consequently, experimental settings in this field often focus on classification problems across multiple conditions, as seen in datasets like ChestXray8.
> > >
> > > Regarding the second point— whether a reconstructed sample is merely an "average-looking chest X-ray" or contains identifiable features from the training data, we acknowledge the validity of this concern. This challenge is further compounded by the inherent complexity of chest X-ray images, which are difficult to interpret accurately without specialized expertise. This is why datasets involving facial images are often preferred in studying model inversion attacks, as they provide clearer identity markers, making the risks and outcomes of such attacks more discernible.
> > >
> > > However, we emphasize that **this challenge is inherent to generative model inversion attacks in general and is not a specific weakness of the approach presented in our work**. While the complexity of certain data types, such as chest X-rays, complicates privacy violation assessments, it does not diminish the importance of studying model inversion attacks. These attacks remain a significant concern, particularly in domains where data have clearer identity markers.
> > >
> > > To address these concerns, **we plan to add a discussion in the manuscript about the scope and applicability of model inversion attacks**, helping readers better understand the contexts in which these methods are most effective for evaluating privacy leakage.
> > >
> > > Lastly, we hope our novel contribution has been clearly articulated in our initial response. If you have any further questions or need additional clarification, please feel free to let us know. We would be happy to discuss and clarify any points further. Thank you again for your thoughtful feedback; we hope this response adequately addresses your concerns.
> > >
> > > [r6] Chen et al. "Knowledge-Enriched Distributional Model Inversion Attacks." In ICCV, 2021.
> > >
> > > [r7] Kahla et al. "Label-Only Model Inversion Attacks via Boundary Repulsion." In CVPR, 2022.
> > >
> > > [r8] Han et al. "Reinforcement Learning-Based Black-Box Model Inversion Attacks." In CVPR, 2023.
> > >
> > > [r9] Nguyen et al. "Label-Only Model Inversion Attacks via Knowledge Transfer." In NeurIPS, 2024.
> > >
> > > [r10] Abdal et al. "Image2stylegan: How to embed images into the stylegan latent space?" In ICCV, 2019.

---

> > > ### Author Response · Authors · 2024-08-12
> > > **A polite reminder about the upcoming discussion deadline**
> > >
> > > Dear Reviewer qx3K,
> > >
> > > As the discussion deadline is approaching, we would like to have a detailed discussion with you to address any new or remaining concerns you may have. We have already provided responses to your previous remaining concerns and would appreciate knowing if they have resolved your issues.
> > >
> > > We look forward to your prompt feedback.
> > >
> > > Best regards,
> > >
> > > Authors of Submission #7116

---

> > > ### Author Response · Authors · 2024-08-13
> > > **Gentle reminder: discussion period concludes today**
> > >
> > > Dear Reviewer qx3K,
> > >
> > > Thank you for your time and valuable comments. We understand you may be quite busy, but the discussion deadline is rapidly approaching.
> > >
> > > Could you kindly review our response and let us know if you have any further questions?
> > >
> > > Thank you for your attention.
> > >
> > > Best regards,
> > >
> > > Authors of Submission #7116

---

> ### Author Response · Authors · 2024-08-11
> **Response to Reviewer qx3K (1/2)**
>
> We're glad to have addressed some of your concerns, and we would like to take this opportunity to further address the remaining points.
>
> > Re: The dynamic prior.
>
> We appreciate your feedback and **understand that the term 'dynamic prior' in our work may have led to some misunderstanding**. To address this, we will revise the manuscript to use a more precise description of our approach. Specifically, we will clarify that we fine-tune the generator $\mathrm{G}$, which represents the prior, during the model inversion process.
>
> The key difference we emphasize is that, in our approach, the generator is continuously fine-tuned throughout the model inversion process. In contrast, previous works such as [r1-r3, r6-r9] select a prior based on a heuristic but do not fine-tune the generator during the model inversion process. This continuous fine-tuning process distinguishes our method. To further clarify the distinction between our approach and previous generative model inversion attacks [r1-r3, r6-r9], we provide the following step-by-step explanation.
>
> In previous generative model inversion attacks, the generative model $\mathrm{G}$ remains **frozen** throughout the model inversion process:
> - Initialize latent codes: $\mathbf{Z}=\\{\mathbf{z\_i} \mid \mathbf{z}\_i \in \mathcal{Z}, i = 1,\ldots, N\\}$;
> - Obtain optimized latent codes: $\hat{\mathbf{Z}}=\\{\hat{\mathbf{z}} = \text{argmin}~\mathcal{L}\_{\text{id}}(\mathbf{z};y,\mathrm{M}, \mathrm{G}) + \lambda \mathcal{L}\_{\text{prior}}(\mathbf{z};\mathrm{G},\mathrm{D}) \mid  \mathbf{z} \in \mathbf{Z}\\}$;
> - Generate recovered samples: $\mathcal{D}_{\text{private}}^{\text{s}} = \\{\hat{\mathbf{x}} = \mathrm{G}(\hat{\mathbf{z}}) \mid \hat{\mathbf{z}} \in \hat{\mathbf{Z}} \\}$.
>
> In contrast, our pseudo-private data guided model inversion attacks (refer to Section 3.2) involve a more dynamic process that includes the following **three iterative steps**:
> - Generate pseudo-private dataset with generative model inversion attacks (**frozen** $\mathrm{G}$): $\mathcal{D}_{\text{private}}^{\text{s}}$;
> - Select high-quality pseudo-private dataset from $\mathcal{D}\_{\text{private}}^{\text{s}}$ (**frozen** $\mathrm{G}$): $\mathcal{D}\_{\text{private}}^{\text{s}'}$;
> - Density enhancement around high-quality pseudo-private data $\mathcal{D}\_{\text{private}}^{\text{s}'}$ (**unfrozen** $\mathrm{G}$): $\mathrm{G}, \mathrm{D} \leftarrow \texttt{Fine-tune}(\mathrm{G}, \mathrm{D}, \mathcal{D}\_{\text{private}}^{\text{s}'})$.
>
> As observed in our approach, the generative model $\mathrm{G}$ is not static but is fine-tuned iteratively based on the selected pseudo-private data. This differs from the static nature of the generator $\mathrm{G}$ in the previous generative model inversion attacks.

---

### Official Review · Reviewer_CEeM · 2024-07-11

**Soundness:** 3
**Presentation:** 2
**Contribution:** 3
**Rating:** 7
**Confidence:** 3

**Summary:**

It is well known that deep learning models are susceptible to model-inversion attacks, which is to say that they can be probed to reveal their training data. The authors design a more powerful method of attack by increasing the density of their prior using “pseudo-private data”, they can increase the probability of sampling actual private data.

**Strengths:**

The contribution of this work is clear and intuitive. The authors provide empirical evidence for their claims and they propose different algorithms which leverage generated samples. They benchmark their method against existing methods and show how they can build on prior work in this area. The illustration and build up to their core result is well structured and makes the paper easy to follow.

**Weaknesses:**

- There are a number of typographic errors (e.g. “vallina”)
- A brief related work section which situates their contribution in the main body would have been appreciated
- There is verbatim repetition in the problem setup.
- They claim that “all state-of-the-art generative MIAs” are limited due to the utilization of a fixed prior during inversion. This bold claim is not backed up.
- Some of the symbols are not properly defined  (e.g. \lambda is undefined)

**Questions:**

- What are the real-world settings that this threat model could be observed with this work?
- How might this approach extend to diffusion-based models?

**Limitations:**

The paper makes the limitations and ethics issues with their work clear.

---

> ### Author Rebuttal · Authors · 2024-08-07
>
> Thank you for your constructive comments and generous supports! Please see our detailed responses to your comments and suggestions below, where we use references in our manuscript due to the token limit.
>
> > W1. There are a number of typographic errors (e.g. “vallina”)
>
> Thank you for your detailed review of our paper and for identifying the typographic errors. We have made the necessary corrections, including changing "vallina" to "vanilla." Your attention to detail is greatly appreciated, and we have carefully reviewed the manuscript to address all identified issues.
>
> > W2. A brief related work section which situates their contribution in the main body would have been appreciated.
>
> Thanks for this suggestion! We have revised the paper to include a brief introduction of related work. Due to space constraints, we integrated this section into the problem setup (i.e., Section 2.1), while the detailed related work is provided in Appendix A.1. The specific content added is as follows:
>
> "Current generative MIAs primarily concentrate on either the initial training process of GANs [Chen et al., 2021, Yuan et al., 2023, Nguyen et al., 2024] or the optimization techniques used in the attacks [Zhang et al., 2020, Wang et al., 2021a, Struppek et al., 2022, Kahla et al., 2022, Nguyen et al., 2023]. In this paper, we take another direction and introduce a novel approach by fine-tuning the GAN’s generator based on the attack results from previous runs. This approach introduces a dynamic and iterative dimension to model inversion attacks, expanding the current understanding and application of generative MIAs."
>
> > W3. There is verbatim repetition in the problem setup.
>
> Thank you for highlighting the verbatim repetition in the problem setup, particularly in the phrases "the goal is to find a sample $\mathbf{x}$ that maximizes the model $\mathrm{M}$'s prediction score for class $y$" and "aimed to optimize for an optimal synthetic sample $\mathbf{x}^*=\mathrm{G}(\mathbf{z}^*)$ to maximize the target model's prediction probability in the target class $y$" in the original main text. We have made the necessary revisions to eliminate the repetition.
>
> > W4. They claim that "all state-of-the-art generative MIAs" are limited due to the utilization of a fixed prior during inversion. This bold claim is not backed up.
>
> Thank you for pointing out the need for further clarification regarding our claim that "all state-of-the-art generative MIAs" are limited due to the utilization of a fixed prior during inversion. We recognize that this statement may seem bold without sufficient supporting evidence.
>
> **However, our intention is not to overgeneralize but to emphasize a prevalent trend observed in multiple state-of-the-art generative MIAs**, which we have validated through rationale-driven analysis and main experiments. To address this concern, we have revised the manuscript by changing "all state-of-the-art generative MIAs" to "state-of-the-art generative MIAs" and have cited specific examples validated in our experiments to demonstrate the common use of fixed priors and their associated limitations.
>
> We appreciate your feedback and will ensure that our claims are accurately represented and well-supported in the revised manuscript.
>
> > W5. Some of the symbols are not properly defined (e.g. \lambda is undefined)
>
> We have carefully reviewed the entire manuscript and added definitions for all previously undefined symbols, including $\lambda$ and $\alpha$. Thank you for bringing this to our attention.
>
> > Q1. What are the real-world settings that this threat model could be observed with this work?
>
> In real-world scenarios, the threat model described in our work can manifest in various situations where sensitive data is vulnerable to exposure through model inversion attacks. A prominent example is in **security and surveillance**. Models deployed in these contexts, such as facial recognition systems, are particularly susceptible to model inversion attacks that could reveal personal identities [r1, r2]. Another critical example is in **healthcare systems**,
> where machine learning models are used to analyze sensitive medical data, such as diagnostic images or patient records. In these settings, an adversary could exploit model inversion techniques to reconstruct confidential information, like detailed diagnostic images (e.g., CT scans), thereby compromising patient privacy [r3].
>
> > Q2. How might this approach extend to diffusion-based models?
>
> To the best of our knowledge, diffusion models have not yet been applied to optimization-based generative model inversion attacks (i.e., the approach outlined in Eq. (1)). We hypothesize that this is primarily due to the technical challenges posed by the multi-step sampling process in diffusion models.
>
> One potential challenge arises in the optimization of the latent code $\mathbf{z}$ during model inversion process, as the denoising Markov chain involves multiple steps. This process requires storing the generator's gradients at each step for backpropagation, leading to substantial memory consumption.  Additionally, errors can accumulate throughout the optimization (i.e., sampling) process, potentially resulting in a suboptimal or inaccurate latent code.
>
> Therefore, while we do not have a definitive answer on how to extend this approach to diffusion models, we believe that diffusion model-based MIAs represent a highly interesting and promising area of research, given the superior generative performance of diffusion models compared to GANs.
>
>
> ---
> **References**:
>
> [r1] Zhang et al.  "The Secret Revealer: Generative Model-inversion Attacks Against Deep Neural Networks." In CVPR, 2020.
>
> [r2] Struppek et al. "Plug & Play Attacks: Towards Robust and Flexible Model Inversion Attacks." In ICML, 2022.
>
> [r3] Wang et al. "Variational Model Inversion Attacks." In NeurIPS, 2021.

---

> > ### Comment · Reviewer_CEeM · 2024-08-12
> >
> > I thank the authors for the clarifications / adjustments and have update my score.

---

> ### Author Response · Authors · 2024-08-13
>
> Dear Reviewer CEeM,
>
> Thank you once again for taking the time to review our paper and for your ongoing support. Your thoughtful feedback has been very valuable in improving our work.
>
> Sincerely,
>
> Authors of Submission #7116

---

### Official Review · Reviewer_4Zsf · 2024-07-14

**Soundness:** 3
**Presentation:** 3
**Contribution:** 2
**Rating:** 5
**Confidence:** 5

**Summary:**

The paper proposes a novel plug-and-play method for current state-of-the-art MI methods to enhance their performance and mitigate the challenge of distribution discrepancy. This method first conducts a round of MI attack to acquire pseudo-private data and then utilizes the data to fine-tune the generative prior following certain strategy. The experimental results show that the proposed method enhances the attacking capability of existing MI methods to some extent.

**Strengths:**

1. Clarity: The paper is well-structured and easy to follow.
2. Originality: The proposed method seems novel and interesting.
3. Intuitive pipeline: The method is easy to understand and implement.

**Weaknesses:**

1. The current state-of-the-art white-box MI method PLGMI [1] should be evaluated.
2. There is no experiment for evaluation on any model inversion defenses. Related experiments are expected to demonstrate how the proposed technique impacts the resistance ability of MI attacks against defense strategies.
3. The middle panel in Fig 5 seems wrong. The attack accuracy of random samples is higher than the one of high-quality samples.
4. In Section 4.3, the citation for Fig 5 appears to be Fig 4.3.

[1]Xiaojian Yuan, Kejiang Chen, Jie Zhang, Weiming Zhang, Nenghai Yu, and Yang Zhang. Pseudo label-guided model inversion attack via conditional generative adversarial network. In AAAI, 2023.
[2]Lukas Struppek, Dominik Hintersdorf, Antonio De Almeida Correia, Antonia Adler, and Kristian Kersting. Plug & play attacks: Towards robust and flexible model inversion attacks. In ICML, 382 2022.

**Questions:**

1. In Section C.4, why is the final results selection stage removed? This technique is critical to PPA [2]. Omitting this stage might leads to the degraded performance of PPA in the Experiment Section.
2. In Section C.4, why the parameters of pre-attack latent code selection stage are significantly lower than the original paper (200 candidates from a search space of 2000/5000 latent codes). This also contributes to the bad performance of PPA [2] in the Experiment Section.

**Limitations:**

The authors have adequately addressed the limitations and potential negative societal impact of their work.

---

> ### Author Rebuttal · Authors · 2024-08-07
>
> Thank you for your time and careful review of our work. Please see our detailed responses to your comments and suggestions below.
>
> > W1, W2: Missing evaluations on PLG-MI and state-of-the-art model inversion defenses.
>
> Regarding these additional evaluations, please refer to the general response. These results will be included in the main results section (i.e., Section 4.2) of the final version of our paper.
>
> > W3. The middle panel in Fig 5 seems wrong. The attack accuracy of random samples is higher than the one of high-quality samples.
> > W4. In Section 4.3, the citation for Fig 5 appears to be Fig 4.3.
>
> Thank you for your careful review of our paper and for pointing out these issues. We have made the necessary corrections. The middle panel in Fig. 5 has been revised to represent the data accurately, and the citation error in Section 4.3 has been corrected to refer to Fig. 5 as intended.
>
> > Q1, Q2: The removal of the final results selection stage and the change of parameters in the pre-attack latent code selection stage.
>
> Our proposed approach, PPDG-MI, requires identity-wise fine-tuning of the generator $\mathrm{G}$ during the model inversion process. Consequently, all three stages in the attack pipeline—latent code sampling, optimization, and final result selection—should be designed to function in an identity-wise manner.
>
> Taking the experiment, where $\mathcal{D}_{\text{public}}$ = FFHQ and $\mathrm{M}$ = ResNet-18, as an example, the original PPA setting involved selecting 200 candidates from 5000 latent codes, running 70 optimization iterations per latent code, and then choosing the 50 recovered samples with the highest average prediction scores as the final results. The time cost we measured for the latent code sampling stage was approximately 52s/identity, the optimization stage took around 565s/identity, and the final result selection stage took about 5s/identity (these measurements were conducted on a single A100 GPU). Therefore, the time overhead is substantial, particularly since our method necessitates at least two rounds of attacks.
>
> Given that latent code sampling in PPA is performed only once, whereas our approach requires identity-wise sampling, we modified the experimental setting by selecting 100 candidates from 500 latent codes to save on experiment time. This also significantly reduced the time required for the optimization stage. In addition, to maximize the amount of test data in our evaluation, we removed the final result selection stage, retaining all 100 optimized latent codes. It is important to note that these setting changes were applied equally to both the baseline PPA and our PPDG-MI, ensuring a fair comparison. Therefore, these adjustments do not affect the validity of our method's effectiveness.
>
> We'd appreciate it if you could consider the above responses when making the final evaluation of our work. Please let us know if you have any outstanding questions.

---

> > ### Comment · Reviewer_4Zsf · 2024-08-11
> >
> > The experimental results are positive. Thanks to the authors for addressing all my concerns and I would like to increase my score.

---

> > > ### Author Response · Authors · 2024-08-12
> > >
> > > Dear Reviewer 4Zsf,
> > >
> > > Thank you for taking the time to review our rebuttal. We are glad to hear that our clarifications and additional results have positively influenced your perspective on our work.
> > >
> > > Best regards,
> > >
> > > Authors of Submission #7116

---

> ### Author Response · Authors · 2024-08-10
> **Would you mind checking our responses and confirming whether you have any further questions?**
>
> Dear Reviewer 4Zsf,
>
> Thanks very much for your time and constructive comments.
>
> Would you mind checking our responses and confirming whether you have any further questions?
> Any comments and discussions are welcome!
>
> Thanks for your attention and best regards.
>
> Authors of Submission #7116

---

### Author Rebuttal · Authors · 2024-08-07

We sincerely thank all reviewers for their thoughtful and insightful suggestions on our submission. We address a few common points in this response. All other questions are addressed in reviewer specific responses.
> Re: The evaluation of PLG-MI [r1].

We have included the experimental comparison with PLG-MI, where we adopted $\mathcal{D}\_{\text{private}}$ = CelebA, $\mathcal{D}\_{\text{public}}$ = FaceScrub, and $\mathrm{M}$ = VGG16. The results are as follows:
| Method           | Acc@1$\uparrow$ | KNN Dist $\downarrow$   |
|------------------|-----------------|------------|
| PLG-MI           | 53.18           | 1450.00     |
| + PPDG-vanilla (ours)    | **65.36**           | **1309.40**     |

The results presented in the table demonstrate that our proposed method significantly improves model inversion performance over PLG-MI.

> Re: The evaluation of state-of-the-art (SOTA) model inversion defenses.


We have extended our evaluation to include state-of-the-art (SOTA) model inversion defense methods BiDO-HSIC [r2] and NegLS [r3]. The experimental setup is as follows:

- For the high-resolution setting, we adopt $\mathcal{D}\_{\text{private}}$ = FaceScrub, $\mathcal{D}\_{\text{public}}$ = FFHQ, and $\mathrm{M}$ = ResNet-152 trained with BiDO-HSIC or NegLS.

- For the low-resolution setting, we use $\mathcal{D}\_{\text{private}}$ = CelebA, $\mathcal{D}\_{\text{public}}$ = CelebA, and $\mathrm{M}$ = VGG16 trained with BiDO-HSIC or NegLS.

We conducted a targeted comparison for each defense model. For high-resolution tasks, we consider PPA; For low-resolution tasks,  we consider KEDMI and LOM. We report top-1 attack accuracy (Acc@1) and KNN distance (KNN Dist) as detailed below:


|                | **PPA**         |                       |
|----------------|-----------------|-----------------------|
| Method         | Acc@1$\uparrow$ | KNN Dist $\downarrow$ |
| No Def.        |   77.85              | 0.8235                      |
| BiDO-HSIC      | 52.50                |0.9546                       |
| + PPDG-PW      | 54.65                |0.9270                       |
| + PPDG-CT      | 57.40                |0.9051                       |
| + PPDG-MMD     | **58.55**                |**0.9017**                      |


|                | **PPA**         |                       |
|----------------|-----------------|-----------------------|
| Method         | Acc@1$\uparrow$ | KNN Dist $\downarrow$ |
| No Def.        |   77.85              | 0.8235                      |
| NegLS          |   11.35              | 1.3051                      |
| + PPDG-PW      |   14.65              | 1.2234                      |
| + PPDG-CT      |   **16.25**              | 1.2233                      |
| + PPDG-MMD     |   13.25              | **1.2187**                      |


|        | **LOM (GMI)**   |                       | **KEDMI**     |                       | **LOM (KEDMI)**   |                       |
|------------------|-----------------|-----------------------|-----------------|-----------------------|-------------------|-----------------------|
| Method     | Acc@1$\uparrow$ | KNN Dist $\downarrow$ | Acc@1$\uparrow$ | KNN Dist $\downarrow$ | Acc@1$\uparrow$   | KNN Dist $\downarrow$ |
| No Def.          |  63.19               |        1416.80               |         75.54        |           1297.79            |        84.10           |           1255.15            |
| BiDO-HSIC        | 47.71          | 1521.50             | 58.50          | 1393.06             | 69.56            | 1420.17             |
| + PPDG-vanilla   | **58.74**      | **1455.31**         | **60.56**      | **1369.28**         | **71.82**        | **1403.60**         |


|                | **LOM (GMI)**      |                       | **KEDMI**        |                       | **LOM (KEDMI)**   |                       |
|----------------|--------------------|-----------------------|------------------|-----------------------|-------------------|-----------------------|
| Method         | Acc@1$\uparrow$    | KNN Dist $\downarrow$ | Acc@1$\uparrow$  | KNN Dist $\downarrow$ | Acc@1$\uparrow$   | KNN Dist $\downarrow$ |
| No Def.        |   63.19                 |      1416.80                 |        75.54          |              1297.79         |        84.10           |           1255.15            |
| NegLS          | 25.40             | 1529.62             | 38.62           | 1335.59             | 69.50            | 1289.03             |
| + PPDG-vanilla | **45.44**         | **1415.76**         | **51.26**       | **1308.22**         | **75.17**        | **1260.65**         |



The experimental results demonstrate that PPDG-MI effectively enhances model inversion performance against models trained with SOTA defense methods.

---
**References**:

[r1] Yuan et al. "Pseudo Label-Guided Model Inversion Attack via Conditional Generative Adversarial Network." In AAAI, 2023.

[r2] Peng et al. "Bilateral dependency optimization: Defending against model-inversion attacks." In KDD, 2022.

[r3] Struppek et al. "Be Careful What You Smooth For: Label Smoothing Can Be a Privacy Shield but Also a Catalyst for Model Inversion Attacks." In ICLR, 2024.

---

### Decision · Program_Chairs · 2024-09-25

**Decision:**

Accept (poster)

**Comment:**

The paper introduces a method to enhancing model inversion attacks (MIAs), which aim to reconstruct class characteristics from a trained classifier. Typically, MIAs rely on training image priors, such as GANs, on public data that differ in distribution from the target model's training data. This distributional discrepancy reduces the effectiveness of MIAs in accurately reconstructing features of the target class. To address this issue, the authors propose pseudo-private data-guided MIAs. The proposed method initially conducts existing MIAs to gather a set of attack samples that reveal features of the target classes. A subset of high-quality samples is selected based on robust prediction scores under the target model. The GAN's generator is then fine-tuned on this subset to increase the sampling density around the pseudo-private data. Subsequent attacks using the updated generator show a significant improvement in attack results across various attacks and settings. However, the reviewers are concerned about the novelty of the work. In addition, the assumptions on the pseudo data distribution are strong, which could lead to bias and model collapse. The paper should further improve the quality based on the reviews.